# Statistically defined visual chunks engage object-based attention

Gábor Lengyel [1,2 ✉], Márton Nagy [1,2,3] & József Fiser [1,2 ✉]

Although objects are the fundamental units of our representation interpreting the environment around us, it is still not clear how we handle and organize the incoming sensory information to form object representations. By utilizing previously well-documented advantages of within-object over across-object information processing, here we test whether learning involuntarily consistent visual statistical properties of stimuli that are free of any traditional segmentation cues might be sufficient to create object-like behavioral effects. Using a visual statistical learning paradigm and measuring efficiency of 3-AFC search and object-based attention, we find that statistically defined and implicitly learned visual chunks bias observers' behavior in subsequent search tasks the same way as objects defined by visual boundaries do. These results suggest that learning consistent statistical contingencies based on the sensory input contributes to the emergence of object representations.

[1] Department of Cognitive Science, Central European University, Budapest, Hungary. [2] Center for Cognitive Computation, Central European University, Budapest, Hungary. [3] Department of Cognitive Psychology, Institute of Psychology, ELTE Eötvös Loránd University, Budapest, Hungary. ✉email: lengyel.gaabor@gmail.com; fiserj@ceu.edu

Instead of perceiving the environment as continuous parallel streams of different information flows, our brain organizes the incoming sensory information into meaningful, distinctive units, called objects, and events determined by causal relationships between these objects. Thus, forming internal representations of objects is fundamental to our perception, and understanding this process is an important step toward developing abstract concepts in the human brain. Yet, it is still unknown what object representations are and how they emerge based on processing and organizing the incoming sensory information.

There is an intensive debate in the field about the cues that are necessary and/or sufficient to form the percept of a visual object dominated by earlier results in object cognition, which demonstrated that stable boundaries defined by luminance contours are one of the strongest criteria for visual "objectness"[1–3]. Indeed, the traditional definition of object representations starts with segmenting the objects from the rest of the input based on boundary information[4–8]. However, similarly to segmenting individual words within a continuous speech during hearing[9–11], segmenting objects from the background is an unresolved challenge in vision as most natural experiences contain ambiguous information about object boundaries leading to a large number of potentially correct segmentations[12,13]. Just as apparent pauses are bad predictors of word endings in speech[14,15], visual edges, contrast transitions, and changes in surface textures are notoriously difficult to identify, and tracking them can lead to false object boundaries[16–18]. In real-life situations, relying exclusively on specific low-level perceptual cues (such as edges) in the sensory input has been proven to be insufficient for finding the true objects in the environment[1,18].

One potential solution to this problem is based on the proposal that it is not edge boundaries that are required for object definitions but instead, they manifest just one (albeit important) example of a more general principle that leads to object representations: consistent statistical properties co-occurring in the input[19]. Such multi-faceted statistical properties might be more ubiquitous, more reliable to detect and, instead of being encoded innately, a large fraction of them can be learned from and tuned by experience similar to how statistical cues help babies to successfully segment speech[20,21]. While this proposal can explain why a wide variety of cues (e.g., disparity[1,22], symmetry[23], or motion[1]) were found to be sufficient to elicit the perception of an object, it has not been systematically evaluated in the past.

To investigate the emergence of object representations and evaluate the relevance of consistent statistical properties in this process, we used the following rationale. If consistent statistical properties acquired by learning are indeed fundamental in forming object representations, then a set of newly learned arbitrary statistical contingencies, even if they are not connected to traditional cues and even if they are learned implicitly, should manifest the same kind of object-based behavioral-cognitive effects as true objects do. To test this hypothesis, we started with an implicit learning paradigm called visual statistical learning (VSL), which uses a set of artificial shape stimuli to create novel scenes (Fig. 1a, VSL - Block 1). Crucially, the only relevant statistical contingencies defining the structure of these scenes are the co-occurrence statistics of the shapes (i.e., the stable shape-pairs in fixed spatial relation that form the scenes) with no link to low-level visual cues[24,25]. Therefore, the low-level contrast edges, texture transitions, or Gestalt structures that can be important in forming classical object boundaries[12,17,26] cannot reveal the statistical structure of the chunks in these scenes. Nevertheless, since these chunks are defined by stable statistical contingencies, according to our hypothesis, they qualify as newly learned objects, and therefore, they should induce object-based perceptual effects.

We measured object-related perceptual effects in our scenes with statistically defined objects in two paradigms. In the first experiment, we designed a novel task following previous studies showing that features within an object are detected better than the same features across two objects[27–31]. We tested whether observers detected a pair of target letters better when they appeared within a chunk than across two chunks that had been learned implicitly in a preceding VSL session. In the second experiment, we used the well-documented object-based attention (OBA) paradigm[32]. This paradigm has been used to show that observers responded faster in a cue-based detection task when the target appeared within the object that a preceding cue had indicated compared to when the target appeared on a previously uncued object[32–36]. We tested whether the same attentional bias would also emerge when instead of objects defined by visual boundaries, the paradigm was applied to newly learned chunks defined by statistical contingencies of abstract shapes. Note that the object-based perceptual effects we measured in these two paradigms were previously attributed exclusively to objects defined by visual boundaries in an explicit manner. Both experiments provided clear evidence that recently and implicitly learned statistical chunks without any visual boundary defined by luminance or other traditional cues elicited the same object-based effects as objects with explicit boundaries did.

## Results

**Experiment 1.** In Experiment 1, we tested whether the internal representation of the statistical structure developed during a standard VSL paradigm[24] could bias the subsequent visual search task similarly to how objects defined by explicit visual boundary cues would. In alternating blocks of VSL and search trials, observers were exposed to a series of scenes composed of abstract shapes (Fig. 1a–d, in the green background). Unbeknownst to the observers, the shape compositions in all the scenes followed a predefined structure based on permanent shape-pairs (Fig. 1a, VSL - Block 1, Inventory). After each VSL block, observers completed a letter-search task with scenes composed of shapes and letters superimposed on shapes (Fig. 1b, Search - Block 1). In each trial, participants had to judge in a three alternative forced-choice (3-AFC) task whether they saw (1) two target letters horizontally arranged next to each other, (2) two target letters vertically arranged on top of each other, or (3) just one target letter. If the shape-pairs (chunks) that could only be learned from the co-occurrence probabilities of the shapes during VSL blocks behave similarly to objects, then the letter search should be facilitated in this setup by the chunks the same way as it would be by contour-based objects. Indeed, we found that observers detected the targets better when they appeared within a chunk than across chunks both in Experiment 1a and in its replication, in Experiment 1b. These results reflected implicit learning processes and not intentional cognitive strategies since we excluded from the analysis participants who gained explicit knowledge of the chunks during the experiment (one participant from Experiment 1b, see Methods for details). Moreover, when we ran a control experiment, Experiment 1c, which was identical to Experiment 1a and b except that we used objects defined by visual boundaries not chunks (Fig. 1e–g, in the red background), we obtained a behavioral pattern in the visual search task, which was very similar to what we found with statistical chunks.

In the first search block of Experiment 1a, the statistical chunks significantly modulated the visual search task. Observers committed more errors when the target letters appeared across chunks compared to when the targets appeared within a chunk ($t_{29} = 4.37$, $p < 0.001$, $d = 0.812$, Bayes Factor = 186, Fig. 2a). After the first block, this effect in the error rate disappeared, none

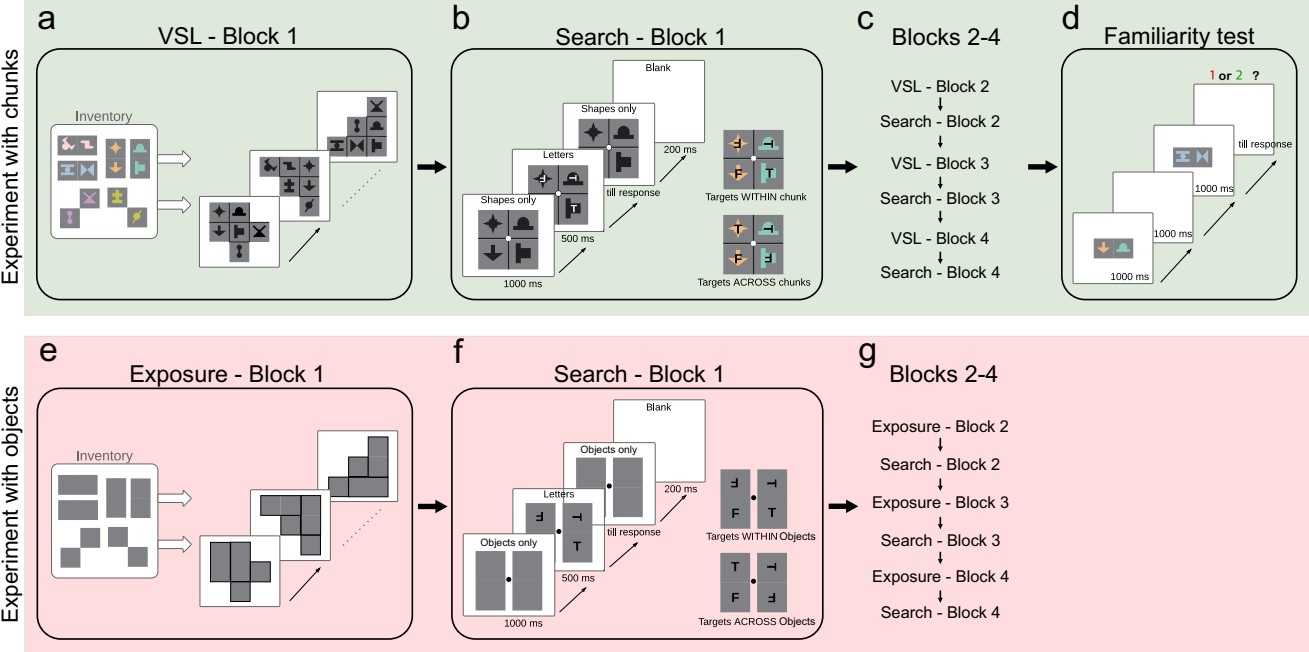

**Fig. 1 The stimuli, the tasks, and the design of Experiment 1. a–d** The design of Experiments 1a and 1b using statistical chunks defined by co-associated abstract shapes. In the Exposure blocks (**a** VSL - Block 1), true-pairs (Inventory) were used to generate 144 complex scenes for passive viewing. In the Search blocks (**b** Search - Block 1), observers performed a letter search task with white letters superimposed on the shapes, where the two target letters could be within or across pairs (**b** inset, using black letters for visibility). Exposure and Search blocks were presented in an alternating manner (**c** Blocks 2–4). After the last Search block, a standard VSL Familiarity test was administered to measure the observer's bias to true chunks over random combinations of elements (**c** Familiarity test). Coloring of the shapes in this figure is only for demonstration purposes, all shapes in the displays were shown in black with no indication of chunk identity. **e–g** The design of Experiment 1c using objects defined by visual boundary cues. In the Exposure blocks (**e** Exposure - Block 1), rectangles and squares were used that corresponded to the silhouettes of the pairs in Experiments 1a and 1b. In the Search blocks (**f** Search - Block 1), observers performed a letter search task with letters appearing in separated rectangles and squares. **f** (insets) Trials within (top) and across (bottom) object setups of targets. The block design of Exp 1c followed that of Experiments 1a and b (**g** Blocks 2–4). The shapes and the letters are magnified in the figure compared to the actual experimental displays.

of the error rate differences in search blocks 2–4 differed significantly from zero ($ts_{29} < 1.91$, $ps > 0.066$, $ds < 0.354$, Bayes Factors < 1, Fig. 2a). The drop in the chunk-based error rate effect between the first and the other three search blocks was also significant ($F_{3,87} = 7.417$, $p < 0.001$, Bayes Factor = 539; post-hoc comparisons of Block 1 vs. Blocks 2–4: $ts_{29} > 3.04$, $ps < 0.004$, $ds > 0.565$, Bayes Factors > 8, Fig. 2a). Meanwhile, there was no difference in the measured reaction times between within-chunk (targets appeared within a chunk) and across chunks (targets appeared across chunks) trials across the four blocks ($ts_{29} < |1.50|$, $ps > 0.145$, $ds < |0.278|$, Bayes Factors < 1, Fig. 2d).

These results indicate that immediately after the first exposure to the novel structured input (1st VSL block), the implicitly learned chunks influenced the accuracy of the observers in the visual search task in the predicted manner: observers detected the two targets more accurately when the target letters appeared within the same statistically defined chunk compared to when they were distributed across two chunks. To confirm that this effect is indeed linked to the implicit learning of the chunks, we calculated the correlation between the chunk-based error rate difference in Block 1 and the amount of statistical learning measured by the Familiarity test, and found a significant effect ($r = 0.40$, $CI_{95} = 0.03$–$0.67$, $p = 0.031$, Bayes Factor = 3, Fig. 2g). This supports the idea that the effect on the error rates was a direct consequence of the learned statistical structures during the VSL block.

The overall performance of the observers did not differ significantly from chance in the Familiarity test ($t_{29} = 1.409$,

$p = 0.169$, $d = 0.262$, Bayes Factor = 0.5, Fig. 2g, orange error bar on the $x$ axis) despite the fact that, in total, observers were exposed to twice as many exposure scenes as in the classic experiment of Fiser and Aslin[24]. The most probable explanation of this is that the scenes in our experiment were divided into four shorter exposure blocks interleaved with the search blocks, and thus the interleaved visual search blocks interfered with the performance in the Familiarity test. Importantly, given the substantial variability in learning found during the Familiarity test, this non-significance of the overall magnitude of learning was irrelevant with respect to the two main results found, namely the differential search behavior of within vs. across learned chunks and the significant correlation between the magnitude of the search difference and statistical learning measured in the Familiarity test.

To enhance the credibility of our results, we reran Experiment 1a with a different group of observers in Experiment 1b. This time, observers completed only two-two blocks of VSL and search trials since in Experiment 1a, the chunks influenced the performance significantly only in the first search block and it disappeared in the remaining of the blocks (Fig. 1a). In the replication Experiment 1b, we obtained exactly the same results as in Experiment 1a (Fig. 2b, e): the chunks had a strong effect on error rates in the first search block ($t_{29} = 2.68$, $p = 0.012$, $d = 0.498$, Bayes Factor = 4), which disappeared in the second search block ($t_{29} = -0.86$, $p = 0.398$, $d = 0.159$, Bayes Factor = 0.3), the chunk-based effect was also significantly smaller in the second block than in the first search block ($t_{29} = 2.41$, $p = 0.022$, $d =$

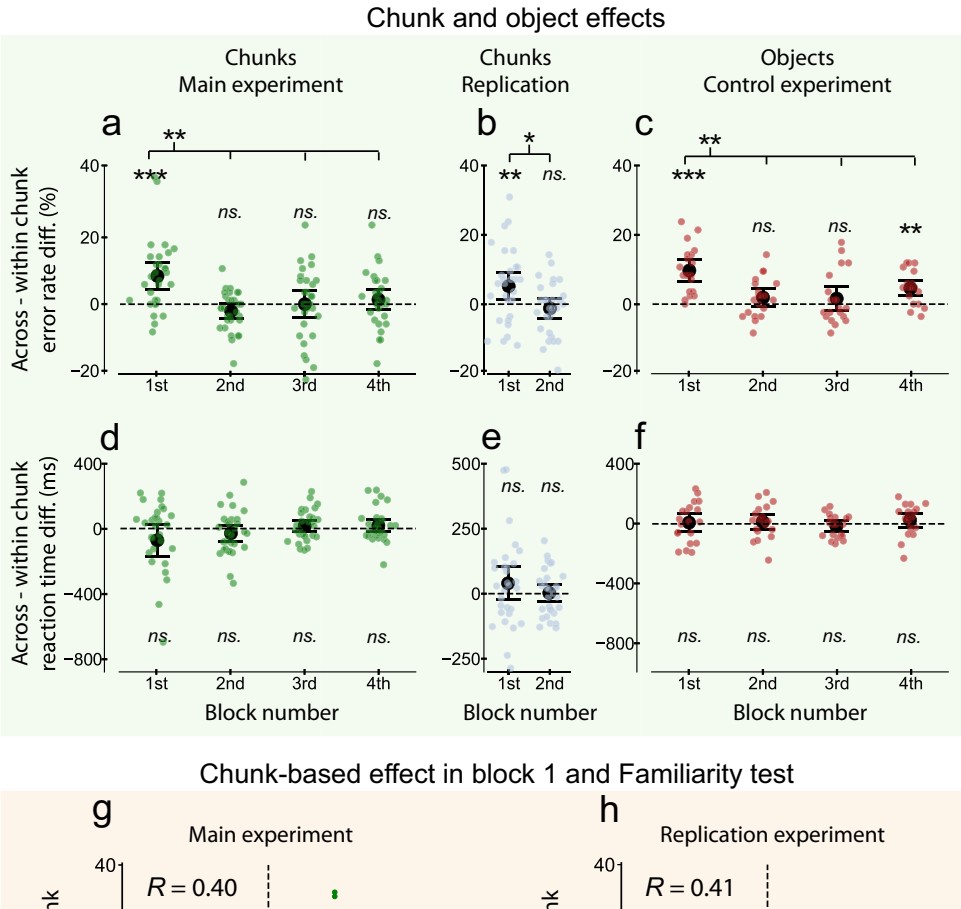

**Fig. 2 Chunk- and object-based error rate effects in Experiment 1. a–c** Chunk/object-based error rate (**a–c**) and reaction time (**d–f**) effects across Exps 1a, 1b, and 1c. Mean error rate and median reaction time differences between the across-chunk and within-chunk trials (y axis) in each Search block (x axis) in the main (**a**) and in the replication (**b**), and in the control (**c**) experiments. Positive values mean fewer errors or faster responses in within-chunk compared to across-chunk trials and error bars show the 95% confidence intervals of the mean. Colored dots represent the mean error rates or median reaction time differences of the observers in a given block. **g, h** The relationship between performance in the Familiarity test (x axis) and error rate differences of the across-chunk vs. within-chunk trials in the first block (y axis) in the main (**g**) and in the replication (**h**) experiments. Green error ellipses show one standard deviation and green lines represent best-fitting linear regression lines. The error bars show the 95% confidence intervals of the mean performance in the Familiarity test (orange), and of the average chunk-based error rate effect (blue). $n = 30$ in Exp. 1a (**a, d, g**), $n = 30$ in Exp. 1b (**b, e, h**), and $n = 20$ in Exp. 1c (**c, f**). Significant differences from zero in **a–f** are indicated with $^{ns.}p > 0.05$, $*p < 0.05$, $**p < 0.01$, $***p < 0.001$, two-tailed paired (the difference between across and within-chunk trials) t-tests. R-values in **g** and **f** indicate Pearson correlation coefficients. Source data are provided in the Source Data file (Fig. 2 worksheet tab in Source Data.xlsx).

0.448, Bayes Factor = 2), and there was no effect of the chunks on the reaction times ($ts_{29} < 1.28$, $ps > 0.212$, $ds < 0.237$, Bayes Factors < 0.4).

Using Bayesian statistics, we could combine the data from Experiments 1a and 1b because the first two blocks were identical in those experiments. We computed the probability of the hypothesis that observers made fewer errors in within-chunk trials than in across-chunk trials (following refs. [37,38]), and found very strong evidence supporting the existence of the chunk-based effect, with Bayes Factor = 2907, indicating that the existence of a chunk-based effect is 2907 times more probable than assuming

no chunk-based effect. Furthermore, the Bayes Factor analysis conducted on the correlations (following ref. [39]) indicated that the probability of an existing positive correlation between the chunk-based effect and the performance in the familiarity test was 24 times more probable than assuming no relationship between the two. These results provided further strong evidence that the chunk-based error rate effect was related to the learned statistical structure.

We conducted two additional tests to further strengthen the assessment that implicit learning of the chunks is the driving force behind the error rate effect, and that the significant positive

correlation between the chunk-based error rate effect and familiarity is not just due to a generic factor such as attention or across-subject variability in overall performance. First, we computed the partial correlation between the performance in the Familiarity test and the chunk-based error rate effect while controlling for the average performance in the task (measured by individual average error rates), and we found significant positive partial correlations in both experiments (Experiment 1: $r = 0.39$, $CI_{95} = 0.03–0.67$, $p = 0.028$, Bayes Factor = 3; Experiment 1b: $r = 0.38$, $CI_{95} = 0.04–0.66$, $p = 0.015$, Bayes Factor = 3). This result further corroborates the idea that the underlying cause of the correlation between the performance in the Familiarity test and the chunk-based error rate effect is the implicitly learned statistical structure. Second, we wanted to rule out the possibility that the chunk-based error rate effect emerged solely because observers paid more general attention to the area of the scene with a true-pair structure compared to the area lacking such a structure due to having just two individual shapes (Supplementary Fig. 4c). To this end, we repeated the analysis on the subset of trials in the search task, which had two true-pairs and no single elements (i.e., two chunks, see Supplementary Fig. 4b). In this case, all positions enjoyed the same advantage from being a part of a chunk, thus the effect had to originate from the targets being within the same chunk. We found the effect of the chunks on the error rate in these trials to have the same size as in the case of the full set (Experiments 1a and 1b together: $t_{59} = 3.77$, $p < 0.001$, $d = 0.487$, Bayes Factor = 64) indicating that the reported chunk-based error rate effect could not be explained by allocating more attention to true-pairs than to individual shapes. In summary, in Experiments 1a and 1b we found convincing evidence that (1) the chunks of the scenes' underlying statistical structure modulated subsequent performance in the visual search task, and (2) this chunk-based error rate effect had a strong positive relationship with the performance in the familiarity test measuring the degree of learning.

However, two additional issues had to be clarified for linking these effects to object representations. First, the effect we found diminished after the first search block, and second, it is unclear exactly how objects with explicit boundaries would influence the same search in the present 3-AFC paradigm. To address both issues, we ran a control experiment (Experiment 1c), which was identical to Experiments 1a and 1b in all aspects except that the underlying scene structure was specified by objects defined by visual boundaries instead of chunks defined by abstract shape-pairs (Fig. 1e–g, in the red background). Comparing the results of Experiments 1a, 1b, and 1c, we found that objects with explicit visual boundaries elicited a very similar pattern of results to those obtained with statistical chunks (Fig. 2c, f and see Supplementary Fig. 2). First, objects with visual boundaries influenced the error rates significantly in the first search block, and also significantly more there than in the rest of the search blocks. Specifically, observers made more errors when the targets appeared across compared to within objects in two of the four search blocks (Block 1: $t_{19} = 6.50$, $p < 0.001$, $d = 1.490$, Bayes Factor = 6237; Block 2: $t_{19} = 1.51$, $p = 0.148$, $d = 0.346$, Bayes Factor = 1; Block 3: $t_{19} = 0.957$, $p = 0.351$, $d = 0.219$, Bayes Factor = 0.3; Block 4: $t_{19} = 4.52$, $p < 0.001$, $d = 1.036$, Bayes Factor = 130, Fig. 2c), but this effect was significantly larger in the first block ($F_{3,57} = 7.709$, $p < 0.001$, Bayes Factor = 441; comparing Block 1 to Blocks 2–4 post-hoc: $ts_{19} > 2.71$, $ps < 0.014$, $ds > 0.621$, Bayes Factors > 4, Fig. 2c). Second, objects with visual boundaries had no modulatory effect on the reaction times in any of the blocks ($ts_{19} < 1.10$, $ps > 0.286$, $ds < 0.252$, Bayes Factors < 0.4, Fig. 2f).

The most parsimonious interpretation of these results is that the reduction of the object/chunk-dependent effect after the first block is due to a floor effect in errors, while the sustained within/

across-object difference in the later block of Experiment 1c is due to the stronger overall effect obtained by using objects with visual boundaries compared to chunk-based objects. In particular, when observers struggle to learn the task, the effect is the largest both for objects and chunks (1st block), while after having learned the task (blocks 2–4), they make, on average, fewer errors, hence the error difference due to the effect of chunks/objects also decreases. Indeed, we found that in all three experiments, observers made the most errors in the first block and after the first block their performance improved significantly (Supplementary material, Experiment 1, Results, Supplementary Fig. 2). Thus, while an overall reduction in error difference occurred across the blocks of all three experiments, due to the stronger modulatory effect of objects with visual boundaries, the within/across-object difference could still be detected in Block 4 of Experiment 1c using the present paradigm, while it became insignificant in Experiments 1a and b.

More importantly, based on the strong effects we found and the quantitative treatment of the diminishing nature of the effect over time, this set of experiments coherently demonstrated in a novel visual search task that statistical chunks learned in a VSL paradigm elicited very similar behavioral effects to those caused by objects defined by clear visual boundaries.

**Experiment 2**. If statistical chunks in a VSL paradigm behave as objects defined by explicit visual boundaries, they should also manifest their effect on attention in classical visual cueing paradigms. To test this conjecture and provide further evidence for similar higher-order effects based on objects with visual boundaries and contingency-based novel statistical chunks, we combined the classic object-based attention (OBA) paradigm with the VSL paradigm in our second experiment. Object-based attention (OBA) is a well-documented example of object-related perceptual effects, which is based on reaction time measurements[32–36]. OBA refers to the phenomenon when observers' attention is drawn to one part of an object and their attention will automatically include the whole object, not just the part singled out by the cue[32,34,36]. In the classic demonstration of OBA, observers are asked to identify a target letter among distractor letters in a two alternatives forced-choice (2-AFC) task after a partially reliable cue indicates where the target might appear in a scene composed of multiple objects defined by visual boundaries. Observers are faster to identify the target if the cue indicates an incorrect location but the location is within the object tagged by the cue as opposed to the situation, when the target appears not only in an uncued location but also in an uncued object even when the distances of the target from the cue are identical in the two conditions (Fig. 3b).

In order to investigate whether the chunks learned in the VSL task elicit an effect similar to OBA, we followed the same design as in Experiment 1. Observers completed alternating blocks of VSL and OBA trials. In the VSL blocks, similarly to Experiment 1, they were exposed to a series of scenes, which were composed of chunks of shapes (Fig. 1a). After each block of VSL, observers completed a set of classical OBA trials[32,35], with one modification: the target and distractor letters appeared superimposed on the shapes of the VSL block, which were arranged in a 2-by-2 configuration (Fig. 3a). After finishing all the VSL and OBA blocks, half of the observers completed an additional four blocks of the classic OBA task, but this time using objects defined by explicit visual boundary cues (Fig. 3b). This arrangement allowed a direct comparison between chunk- and contour-driven OBA within these observers. Finally, all observers completed a Familiarity test with chunks. We found that observers identified the targets faster when they appeared at an uncued location which

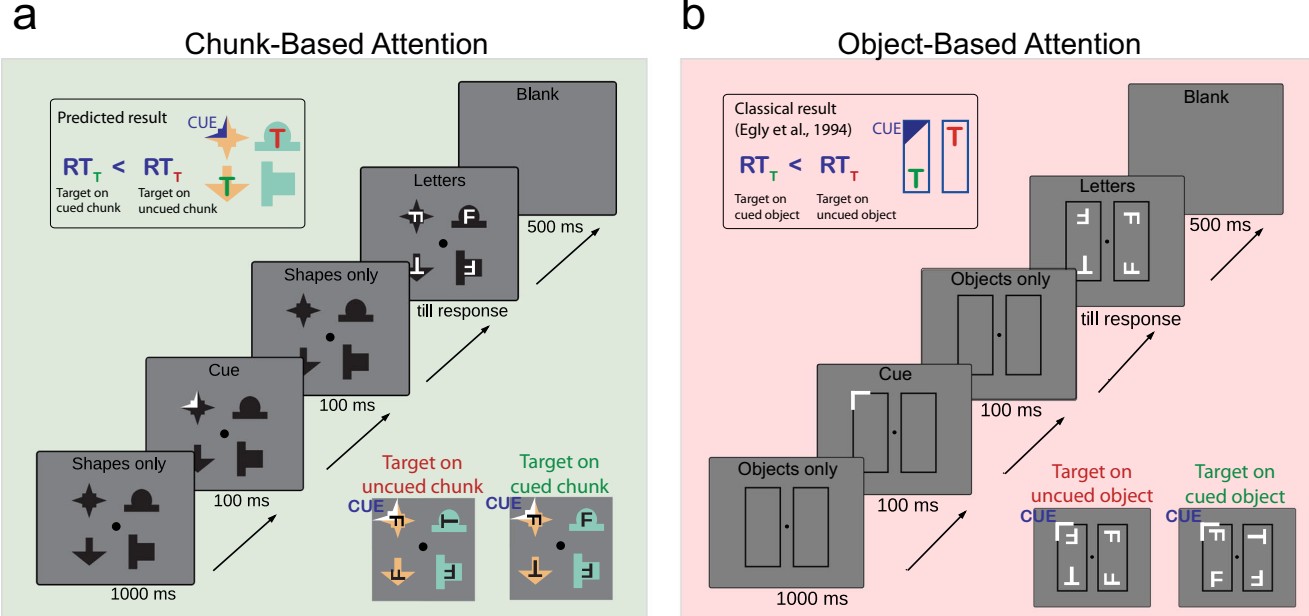

**Fig. 3 The stimuli, the tasks, and the trial structures in Experiment 2. a** Chunk-based attention paradigm. **b** Object-based attention paradigm. **a, b** Top-left insets display the expected results in the two paradigms (longer RTs when the target appears on the uncued chunk/object vs. cued chunk/object). Bottom-right insets in **a, b** present two examples of trials, in which the target (T) appeared on the cued (green label) and on the uncued (red label) chunks/objects. The design, the visual statistical learning, and the Familiarity test were identical to Exp. 1 (Fig. 1). The shapes and the letters are magnified in the figure compared to the actual experimental displays.

was on a cued chunk compared to when the target was presented on an uncued chunk, replicating the exact same pattern that was found with objects defined by visual boundaries. Furthermore, we found a positive correlation between the OBA and the chunk-based attention (CBA) effects in observers performing both tasks, which suggests overlapping cognitive mechanisms behind the two effects. We also found a strong correlation between the CBA effect and the strength of chunk learning as quantified by the Familiarity test. Similar to Experiment 1, we excluded from these analyses all participants with explicit knowledge about the chunks (5 participants, see "Methods" section) to assure that our results reflect the consequences of implicit statistical learning and not explicit strategies.

First, we assessed the standard cue validity effect by measuring how much observers' reaction times and error rates were modulated when the cue indicated the subsequent target position exactly. We found in both the chunk and the object version of the paradigm that observers responded faster (Objects: $t_{43} = 9.78$, $p < 0.001$, $d = 1.491$, Bayes Factor $= 5·10^9$; Chunks: $t_{89} = 11.35$, $p < 0.001$, $d = 1.203$, Bayes Factor $= 10^{16}$; Fig. 4a, left panel), and they made fewer errors (Objects: $t_{43} = 2.46$, $p = 0.018$, $d = 0.375$, Bayes Factor $= 2$; Chunks: $t_{89} = 4.11$, $p < 0.001$, $d = 0.435$, Bayes Factor $= 217$; Fig. 4a, right panel) when the target appeared at the cued (valid-cue trials) compared to the uncued location (invalid-cue trials). There was no difference between the magnitude of the validity effect in the object vs. the chunk version of the paradigm (reaction times: $t_{86} = 0.81$, $p = 0.418$, $d = 0.178$ Bayes Factor $= 0.3$; error rates: $t_{86} = 0.49$, $p = 0.627$, $d = 0.106$, Bayes Factor $= 0.2$; Fig. 4a). Furthermore, there was a large positive correlation between the validity effects using objects and chunks ($r = 0.63$, $CI_{95} = 0.40$-$0.78$, $p < 0.001$, Bayes Factor $= 3499$; Fig. 4b) suggesting that observers who produced a large validity effect in the chunk version also produced a large validity effect in the object version of the paradigm. These results confirm that classical cueing worked in a very similar manner with objects and chunks.

Beyond cue validity, we also successfully replicated the OBA effects reported in earlier studies using objects with visual boundaries[29,32,33,35]. In the invalid-cue trials, observers responded faster when the target appeared in the cued object albeit not in the cued position (cued-object trials) compared to when it appeared in the uncued object (uncued-object trials) demonstrating the classic OBA effect ($t_{43} = 6.62$, $p < 0.001$, $d = 1.010$, Bayes Factor $= 3·10^5$, Fig. 4c, left panel, in red). More importantly, we found the same pattern of results when statistically defined chunks were used instead of objects with clear boundaries. Observers identified the target faster when it appeared on the cued chunk (cued-chunk trials) compared to when it appeared on the uncued chunk (uncued-chunk trials) demonstrating a clear CBA effect ($t_{89} = 2.58$, $p = 0.011$, $d = 0.273$, Bayes Factor $= 3$, Fig. 4c, left panel, in blue). We expected the CBA effect to be smaller than the OBA effect because the former effect emerges due to chunks implicitly learned in the last half an hour while the latter effect is due to objects based on lifelong learning of visual boundary cues. Indeed, the CBA effect was significantly smaller than the OBA effect ($t_{43} = 3.84$, $p < 0.001$, $d = 0.586$, Bayes Factor $= 68$, Fig. 4c). However, there was a significant positive correlation between the CBA and OBA effects ($r = 0.33$, $CI_{95} = 0.03$-$0.58$, $p = 0.026$, Bayes Factor $= 3$, Fig. 4d) providing substantial evidence towards a positive relationship between chunk- and object-based attention. A further link could be established between cue validity and OBA by comparing the results in Fig. 4b, d. The cue validity effect in Fig. 4b indicates the correlation between object- and chunk-based effects for trials where the cue predicted exactly where the target would appear, whereas Fig. 4d shows the same correlation for trials where the cue indicates only the correct object/chunk, but not the correct location. The correlation of $r = 0.63$, obtained in the former case, where the object and chunk-based cueing conditions are highly similar, puts an upper bound on how strong the correlation could be in the latter case had the two processes shared exactly the same underlying mechanism. Therefore, the $r$

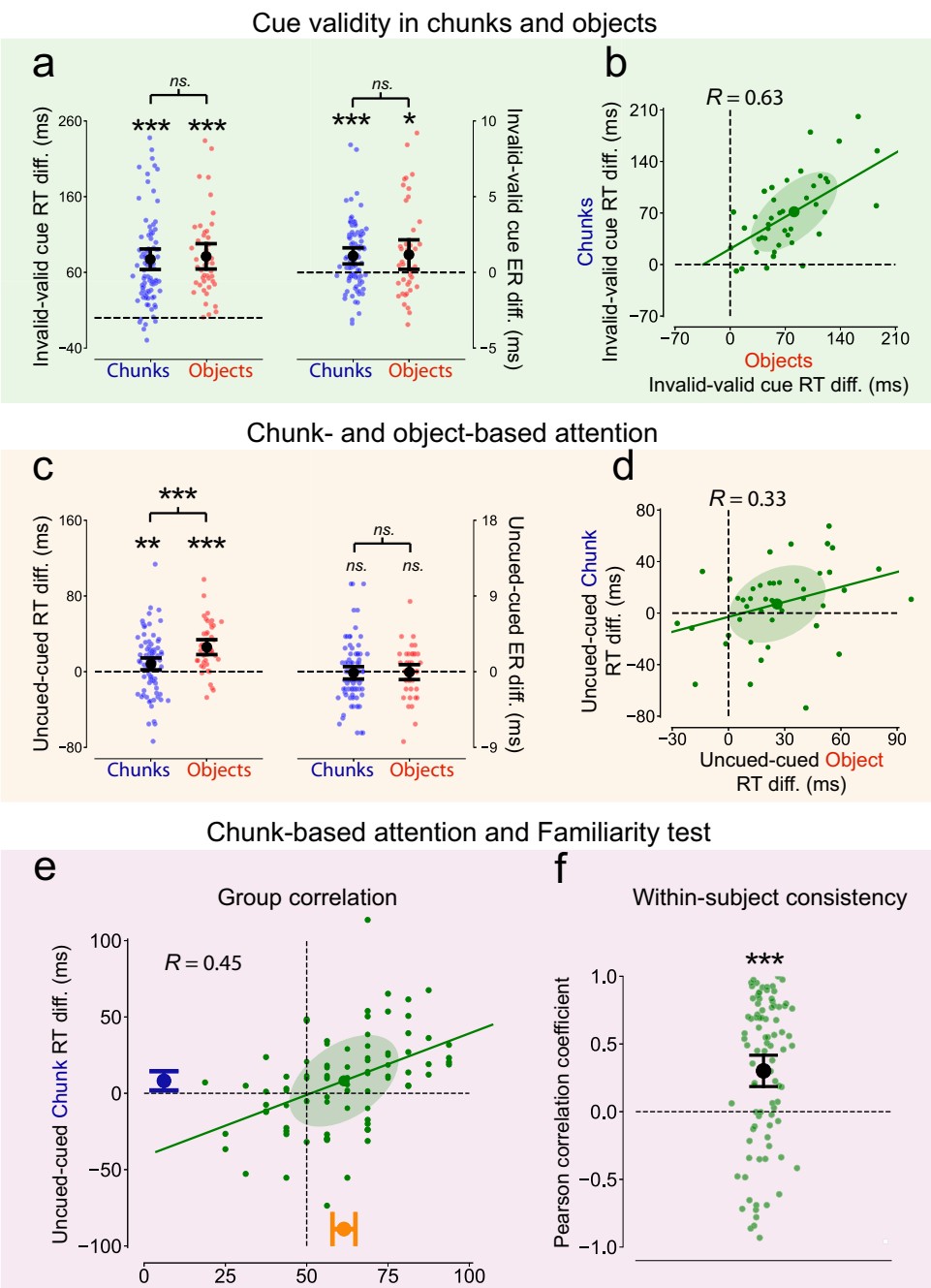

**Fig. 4 Chunk- and object-based attentional effects in Experiment 2. a** The cue-validity effect for chunks (blue) and objects (red). Dots represent the individual observers' validity effect defined as the difference between the median reaction times (right) and mean error rates (left) in the invalid- (uncued) and valid-cue (cued) trials. **b** Correlation between object-based (x axis) and chunk-based (y axis) cue validity with dots representing the corresponding validity effect for each observer. **c** The chunk-based (CBA, blue) and object-based attention (OBA, red) effects. Dots represent the individual observers' OBA/CBA effect defined as the difference between the median reaction times (right) and mean error rates (left) in trials with the target being in an uncued vs. cued chunk/object. **d** Correlation between object-based (x axis) and chunk-based (y axis) attention effects on reaction times with dots representing the corresponding attention effect for each observer. **e** Correlation between the learned statistical structure and the CBA effect with dots in the scatter plot representing each observer's percent correct values in the Familiarity test (x axis, mean in orange) and the extent of their CBA effect (y axis, mean in blue). **f** Within-subject consistency between learning chunks and the evoked CBA effect. Green dots represent the observer's Pearson correlation coefficient between their fraction correct scores and the extent of the CBA effect for each individual chunk. In all plots, error bars denote the 95% confidence intervals of the mean, error ellipses cover one standard deviation, and solid lines represent best-fitting linear regression lines. In the axis labels, RT stands for reaction time and ER stands for error rate. $n = 90$ in the blocks with statistical chunks (**a**, **c**, **e**, **f** in blue and green), and $n = 44$ in the blocks with geometric objects (**a**–**d** in red and green). Significant differences from zero in **a**, **c**, and **f** are indicated with [ns.]$p > 0.05$, *$p < 0.05$, **$p < 0.01$, ***$p < 0.001$, two-tailed paired (difference between uncued and cued or invalid and valid chunk/object trials) t-tests. R-values in **b**, **d**, and **e** indicate Pearson correlation coefficients. Source data are provided in the Source Data file (Fig. 4 worksheet tab in Source Data.xlsx).

$= 0.33$ obtained in Fig. 4d suggests that chunks and contour-based objects evoke significantly overlapping cognitive processes. There were no similar effects in the error rates either for trials with objects or with chunks (Objects: $t_{43} = -0.16$, $p = 0.872$, $d = 0.025$, Bayes Factor $= 0.2$; Chunks: $t_{89} = -0.42$, $p = 0.671$, $d = 0.045$, Bayes Factor $= 0.1$; Fig. 4c, right panel).

Next, we tested whether our CBA effect was not just a spurious finding. We found a very significant positive correlation between observers' performance in the Familiarity test, which indicated the extent of their learning, and the size of their CBA biases ($r = 0.45$, $CI_{95} = 0.26$-$0.61$, $p < 0.001$, Bayes Factor $= 2833$, Fig. 4e). To confirm that this strong positive relationship between learning and CBA was not merely due to changes in generic (e.g., alertness-based) processes, we conducted a within-subject consistency analysis. For each observer, we measured how much the particular chunks they preferred more strongly during the Familiarity test were also the ones that elicited a larger CBA effect. Comparing Familiarity scores and CBA effects for each observer and each chunk separately, we found a very strong and significant within-subject consistency ($r = 0.27 \pm 0.06$, $t_{89} = 4.53$, $p < 0.001$, Bayes Factor $= 941$; Fig. 4f).

Finally, as in Experiment 1, we measured the CBA effect in the trials in which only two true-pairs were presented to rule out the possibility that the CBA effect emerged only in trials with individual shapes because participants allocated more attention to the true-pairs than to the two individual shapes (Supplementary Fig. 4). We found that the CBA effect was detectable in trials with two true-pairs, and it was significant with the same effect size ($t_{89} = 2.57$, $p = 0.012$, $d = 0.273$, Bayes Factor $= 3$). This again indicates that the chunk-based error rate effect cannot be explained by allocating more attention to true-pairs than to individual shapes per se.

Taken together these results, the chunks learned during VSL elicited a very similar attentional effect to what objects with explicit visual boundaries are known to generate. Furthermore, this chunk-based effect was strongly related to the implicitly learned statistical structure during the VSL, since the stronger a chunk was preferred in the Familiarity test, the stronger attentional effect it evoked in the CBA paradigm. Finally, the correlation between CBA and OBA suggests that related mechanisms could be involved when processing objects or chunks supporting the claim that statistical learning creates object-like representations.

## Discussion

The present study provides the first evidence that statistically defined chunks influence visual processes in subsequent search tasks the same way as objects defined by articulated boundary cues do. In the first experiment, observers performed better in a novel 3-AFC visual search task when the targets appeared on the same chunk as opposed to when the targets appeared on two different chunks. In the second experiment, chunks elicited the same object-based attention effect as was reported in the classical findings of Egly et al.[32]. In both experiments, the chunk-based effect was larger in observers who performed better in the familiarity test that measures the observers' implicit knowledge of the statistical structure embedded in the stimuli. These results have implications in two domains of the research on internal representation in the brain: the nature of object representation and the role of learning in having object representations.

Object representation initially has been approached as a boundary contour problem[4,40] that later evolved into characterizing a large number of important cues for object formation, such as good continuation[41,42], closure[43], connectedness[3], convexity[44,45], and regularity of shape[23,46]. Here, we argue for a parsimonious integration of these results by stating that the notion of boundary information for the brain is more general than edge contours, and it is based on separating two sets of consistent elements according to some complex statistical measure, which naturally leads to object representations. In the simplest case, these are dark and light local regions giving rise to a luminance boundary or edge. However, apart from such first-order boundaries, there exist for example second-order boundaries that are invisible to mechanisms detecting first-order boundaries, do not necessarily co-occur with the first-order boundaries, and have ecological relevance[47,48]. In addition, there are texture-based, disparity-based, or motion-based boundaries[22] that can be largely independent of luminance-based boundaries and that are more difficult to perceive without prior experience. In this ordering of increasingly abstract examples of boundaries, discontinuities in any arbitrary measure of the stimulus detected by mid-level routines, or boundaries defined by Gestalt rules are at an even higher level, while the stimuli used in our study reside at the opposite extreme from edges: our elements are grouped and separated based on purely statistical consistencies of co-occurrence without the use of any other low-level visual measure. Yet they evoke the same treatment by our cognition as true contour-based object stimuli do even if only to a smaller extent. Thus, we propose that object representations are defined and object-based effects emerge whenever a sufficient subset of statistical contingencies at various levels of abstraction together indicate a separable entity. We also propose that although objectness seems to be an all-or-none property in most natural settings, in fact, it is a continuum with different degrees of objectness. For example, two solid objects separated by a clear visual gap are perceived as two separate objects until they start to move coherently[2], when they are interpreted as one object with two parts or with a surface marker, and the degree of perceived single-objectness will depend on the level of motion coherence between the two objects.

Regarding the role of learning in forming boundaries and objects by statistical contingencies, a number of earlier results corroborate our proposal that statistical learning leads to object-like representations. In a recent study using a similar VSL paradigm, the observer immediately and automatically generalized between the haptic and visual statistical definition of an object from unimodal experience[19] suggesting that learning statistical chunks in any one modality automatically creates abstract and amodal representations. Several findings suggest that VSL interferes with perception: it affects the extraction of summary statistics of scenes[49], automatically biases attention[50], modulates perceived numerosity[51], creates novel object associations based on transitive relations[52], and influences the size perception of the elements within the structure[53]. Two earlier studies linked perceptual organization and statistical learning between abstract shapes directly[54,55]. In Vickery and Jiang[54] chunks were explicitly delineated from the surrounding with a clear black line, and they found that learning new shape associations with such explicit visual cues led to perceptual grouping. Zhao and colleagues[55] showed that detecting color change was faster within than across chunks that were defined solely by co-occurrence statistics. Unlike in our paper, observers in that study completed the Familiarity test, in which the true chunks were explicitly shown, before the color change detection task with the chunks, and therefore, they had an explicit memory of the underlying chunks. Nevertheless, these studies provide partial support to our claim that statistical learning has a key role in the emergence of object representations in humans.

Another support for the crucial role of learning in forming object representations comes from infant studies. Automatic VSL has been demonstrated amply across various modalities not only

in adults but in infants as well[56–58], suggesting that infants and adults are equally capable of learning the co-occurrence statistics of scenes[9]. Infants are also known to segment and represent objects initially only by a subset of the available sensory cues, the most important cues being surface motion and arrangement, while their ability to utilize the other cues, such as Gestalt rules or smooth contours develops later[1]. This gradual incorporation of more complex cues by infants[1,59] is compatible with the idea that statistical learning mechanisms have a key role in the emergence and elaboration of object representation during infancy. Further support comes from another line of infant studies demonstrating that prior experience with given objects together or separately brings forward the time when the infant is able to perform object segregation properly with the particular objects[60–62]. While these results are strongly suggestive, future investigations will be required to test precisely the relative importance and limits of statistically learned vs. innately available cues in object representation across ages.

Our results show only a correlation between the measured object-based effects and the amount of learning, thus we cannot completely exclude the possibility that the co-variation is due to a common source, and learning contingencies is not causally linked to the emergence of object-like representation of the input. An alternative interpretation of our results could be that object-based attention is not really object-based, and objects and chunks share this kind of attentional effect, which should be properly called "object-and-chunk-based" attention. However, this is unlikely for two reasons. First, the correlation remained strong after controlling general improvement in performance, and this reduces the probability of an uncovered common cause since assuming a dynamically strengthening hidden cause that is related neither to general performance nor to learning contingencies is implausible. Second, there exists no visual cue in our chunk stimuli other than statistical contingencies that would selectively map to the features that were implied as causes of OBA in objects, while the features that were implied (long contours, similar textures/colors, Gestalt structures, etc.) all represent strong examples of statistical contingencies. Therefore, based on parsimony, we propose that the emergence of the chunk-based advantage in Experiment 1 and the chunk-based attention in Experiment 2 are direct consequences of implicitly collecting enough statistical evidence by VSL to treat the chunks as a preliminary object, and automatically initiating object-related processes on them. Clearly, this does not mean that the object-like representation emerging after a brief VSL can be considered as fully-blown, real mental objects, as these preliminary object-like representations need to be fortified by further experiences to pass several additional criteria to reach the representational richness of true mental objects. Whether and under what conditions VSL mechanisms can produce such fully developed mental object representations needs to be clarified by future studies.

Earlier computational studies can point to possible computations showing how statistically defined chunks and objects are related[63–65]. When observers are faced with an unfamiliar environment with an unknown statistical structure composed of shapes, they learn and compress the information about the stimuli in terms of meaningful latent chunks from the shapes instead of representing only recursive pairwise associations between those shapes[63,64]. Therefore, we argue that these latent chunks extracted hierarchically based on the statistical regularity in the sensory input are the building blocks of object-like representations. Investigating visual scenes with low-level features, a recent study provided a computational framework, based on hierarchical Bayesian clustering, that demonstrated how an image can be represented by mixture components organized hierarchically, and how such representations can capture most

Gestalt rules through probabilistic inferences[66]. Such hierarchical chunk-representations, using probabilistic learning, that makes inferences across multiple levels simultaneously can also link VSL-and, therefore, object representations- to low-level perceptual effect and perceptual learning[65].

Regarding the neural correlates of object-based perceptual effects, an fMRI study reported that in the early visual cortex, visual error predictions spread between the parts of the same object[67]. This suggests that already in the early visual cortex, the context for computing the prediction error is defined by the objects rather than by low-level visual cues. If this is correct, early visual areas should also manifest increased gamma synchrony with higher areas similar to what has been reported in relation to object-based attentional effects between the inferior frontal junction and the fusiform face and parahippocampal place areas[68]. Moreover, we posit that this effect should increase with learning the underlying chunk-structure of an unknown visual stimulus.

In conclusion, the present results provide a significant step toward linking the concept of object representations to implicit statistical learning of environmental structures through redefining the fundamental requisites necessary for the perception of a new object.

## Methods

### Experiment 1

*Participants.* Eighty-one university students (53 female, mean age = 21, range = 18–29, 71 right-handed, 49 had normal vision without correction) gave informed consent prior to participation in the experiment. Thirty participants took part in Experiment 1a, 31 in Experiment 1b (replication), and 20 in Experiment 1c (control). We excluded one participant from Experiment 1b, who explicitly noticed the statistical structure of the pairs (s/he could recall the pairs and the shapes consisting of the pairs during the debriefing, see "Debriefing" section) since we were interested in the effects of implicit automatic processes and not the consequence of explicit cognitive knowledge. All participants had a normal or corrected-to-normal vision. The experimental protocols were approved by the Ethics Committee for Hungarian Psychological Research.

*Stimuli.* Similar to previous studies[24,69], the stimuli in the visual statistical learning (VSL) and search blocks in Exps 1a–1b consisted of 12 moderately complex 2D abstract shapes (Fig. 1a, VSL - Block 1, Inventory). Unbeknownst to the observers, an Inventory of 6 pairs was constructed from these shapes creating two horizontally, two vertically, and two diagonally oriented pairs. These pairs were the building blocks of the scenes throughout the experiments, as the two elements of a given pair always appeared together in the prespecified spatial configuration defined by the Inventory. Hence, each pair constituted one statistical chunk in our experiment. For each observer, the shapes were randomly reassigned to the pairs in the inventory to eliminate any specific learning effect across subjects due to particular shape combinations.

*Tasks and procedure.* Observers completed 4-4 (in Experiments 1a and 1c) and 2-2 (in Experiment 1b) alternating blocks of VSL and Search trials. Both Experiments 1a–1b were completed with a final Familiarity test and a debriefing, whereas in Experiment 1c such a Familiarity test was omitted as it was not meaningful (Fig. 1).

*Visual statistical learning paradigm.* Observers watched a series of scenes, each constructed from three pairs chosen pseudo randomly from the Inventory (Fig. 1a, VSL - Block 1). In each scene, one pair was selected from each of the three types (horizontal, vertical, and diagonal). The three selected pairs could appear in a 3-by-3 grid and their positions were randomized with the constraint that each pair had to be adjacent by side to at least one other pair. This method yielded 144 unique scenes with each pair appearing 72 times during each VSL block. We split the possible scenes into two sets so that each pair appeared in each set 36 times, and presented the two sets alternating: the first set was presented in blocks 1 and 3, while the second set in blocks 2 and 4, all in a different randomized order across observers. Each scene was presented for 2 s with 1 s pause between scenes. The task of the participants was simply to observe the scenes passively so that they could answer some questions related to their experience afterward.

*Visual search paradigm.* After each VSL block, observers had to complete a block of search trials. In these blocks, four shapes were presented in each trial adjacent to each other in a 2-by-2 arrangement (Fig. 1b, Search - Block 1). The scenes could contain two true pairs (the two horizontal or two vertical pairs of the Inventory), or one true pair and two individual shapes chosen randomly from the remaining

shapes of the two diagonal pairs. The chunks of diagonal pairs were sacrificed in the search task in order to get more possible unique scenes with a 2-by-2 configuration. In each search block, we presented 144 scenes in random order, from which there were 96 unique scenes containing one true pair and two individual shapes (from the two diagonal pairs) and 12 × 4 unique scenes consisting of two true pairs (the horizontal and vertical pairs). All individual shapes were presented 48 times during the search blocks and all horizontal/vertical pairs were presented an equal number of times.

In each search trial, a small white letter appeared in the middle of each black shape, which could be either a T or an F. The task of the observers was to look for the letter Ts among distractor letters (F), and in a 3-AFC task, they had to press 1 on the keyboard if they saw two Ts horizontally arranged next to each other, press 2 if they saw two Ts vertically arranged on top of each other, and press 3 if they saw only one T. The response key mapping (1-beside, 2-top, 3-one target) and the target letter identity (T or F) was counterbalanced across observers. The letters appeared for 500 ms, then they disappeared and only the shapes were visible until the response (Fig. 2b, Search - Block 1). Observers were instructed to always keep their eye on the fixation dot in the middle of the scene. When two Ts appeared, they formulated either a horizontal or a vertical pair, and these pairs were randomly distributed the same number of times across the four possible locations in the 2-by-2 configuration. Similarly, when only one T appeared in a trial, its position was randomly and evenly distributed across the four possible locations (top-left, top-right, bottom-left, bottom-right). Each of the three response types (targets on top of each other, targets beside each other, only one target) occurred 48 times randomly distributed across the block.

*Familiarity test.* After the last search block of Experiments 1a and 1b (replication), observers completed a 2-AFC task typically administered in VSL experiments. In each trial, they saw two pairs of shapes after each other, and they decided which of the two consecutive pairs seemed more familiar to them based on the experiment (Fig. 1d, Familiarity test). The two pairs were presented sequentially for 1 s each with 1 s pause between them. One of the pairs was a true pair (a horizontal or a vertical pair were chosen from the Inventory; Fig. 1a, VSL - Block 1, Inventory, top four pairs), while the other random pair was constructed from two shapes arbitrarily chosen from the diagonal pairs (Fig. 1a, VSL - Block 1, Inventory, bottom two pairs). Observers performed 8 trials, in which one of the horizontal and one of the vertical true pairs were chosen randomly and tested twice against two randomly paired shapes from the diagonal pairs. The presentation order of the true pair and the random pair was counterbalanced across trials, and the presentation order of the trials was randomized individually for each observer.

*Debriefing.* VSL is an implicit learning task because observers had no task to perform beyond paying attention to the scenes. However, their knowledge of the statistical structure, that they built during the implicit learning task, could become explicit. Since the Familiarity test does not indicate to what extent the responses were based on implicit or explicit knowledge, we conducted a debriefing at the end of Experiments 1a, 1b, and Experiment 2 (see "Methods" section, Experiment 2) to identify observers with clear explicit knowledge of the statistical structure. Participants were questioned whether they noticed anything about the shapes during the experiment. If they answered "yes", they were asked further about what they noticed, and if they said something about pairs of shapes being linked, they were asked to name the shapes in each pair that they remembered. Observers who mentioned noticing consistent pairs during the experiments were considered to be explicit learners who were aware of the hidden statistical structure and, therefore their data were excluded from the analysis.

*Control experiment.* The control experiment, Experiment 1c was identical to Experiments 1a and 1b with the exception that instead of shape-pairs, geometric objects defined by explicit visual boundaries were used as inventory elements (Fig. 1e, in the red background) and there was no Familiarity test at the end. We used rectangles to represent the true horizontal and vertical pairs, and two squares to represent the two constituent shapes of each diagonal pair. In the Exposure blocks, observers saw the same number of scenes constructed from the same constituents in the same manner as in the scenes of Experiments 1a–1b, but constructed by rectangles and squares instead of the pairs of shapes. Consequently, the global silhouettes of the composed scenes were also identical across the three experiments. The Search blocks were as similar to those in Experiments 1a–1b as possible. Observers completed the same number and type of trials with the same target locations as in the first two experiments: either two horizontal or two vertical rectangles, similarly to trials with two true pairs in Exps 1a–1b, or one rectangle and two squares, similarly to trials with one true pair and two individual shapes.

## Experiment 2

*Participants.* We estimated the effect size of the original object-based attention (OBA) reported in previous studies and found that, on average, OBA has a small effect size (Cohen's $d = 0.22$). Since chunk-based attention (CBA) is likely to be even weaker than OBA, we assumed that CBA would yield an effect half as strong as in OBA. Asking for a 60% probability to find the CBA, we established that our study required a sample size of 104 observers. We aimed at one hundred observers and managed to recruit 98 university students (68 female, mean age = 21, range =

18–26, 91 right-handed, 60 had normal vision without correction), who gave informed consent prior to participation in the experiments. As in Experiment 1, observers with explicit knowledge of the chunks were excluded (5/98). We excluded three additional observers because they did not finish the experiment, thus data of 90/98 observers were analyzed in this experiment. All observers had a normal or corrected-to-normal vision. The experimental protocols were approved by the Ethics Committee for Hungarian Psychological Research.

*Stimuli, tasks, and procedure.* The design of the experiment, the stimuli, the VSL blocks, and the familiarity test were identical to Experiment 1 with the exceptions specified below. Observers completed 4-4 alternating blocks of VSL and CBA trials. Due to data acquisition error, 19 observers completed only 3-3 blocks of VSL and CBA trials, but this only reduced the number of trials to 72 from 96 in the experimental conditions, thus their data was used in the analyses. All observers completed a Familiarity test at the end of the final CBA block.

*Visual statistical learning paradigm.* Based on the assumption that stronger associations lead to a larger effect in the CBA task, the number of exposure scenes in the VSL blocks was doubled from 72 to 144 to strengthen the learned associations between the shapes. Set size of 144 was chosen to have a robust learning effect while avoiding an explicit understanding of the input structure. In contrast to the exposure scenes of Experiment 1, the black lines separating the shapes in the scene were completely omitted in order to further decrease lower-level visual cues of structure.

*Chunk-based attention paradigm.* After each VSL block, observers completed a block of CBA trials. In the CBA blocks, four shapes were presented adjacent to each other in a 2-by-2 arrangement without explicit black lines separating them (Fig. 3a). The configurations of the different scenes were the same as in Experiment 1: they either contained two horizontal or two vertical true pairs (Fig. 1a, VSL - Block 1, Inventory, top four pairs, see also Supplementary Fig. 4b) or one true pair and two individual shapes from the diagonal pairs (Fig. 1a, VSL - Block 1, Inventory, bottom two pairs, see also Supplementary Fig. 4c). Observers were exposed to the same number and mix of scenes as in Exp. 1, and the scenes were presented in a different random order in each search block.

Following the original OBA method[32] in each trial scene, first, only the four shapes appeared for 1000 ms, then one of the shapes was cued for 100 ms. The cue disappeared and only the four shapes were visible for another 100 ms, then one target (T or L) and three distractor letters (F) appeared, one in the middle of each of the four shapes. The letters remained in the center of the shapes until the observer responded. Cueing was provided by coloring a quadrant of the black shape to white (Fig. 3a, CBA panel insets). The cue-coloring was designed to draw attention without favoring direction to any location. The observers' 2-AFC task was to press 1 when they saw a letter T, and 2 when they saw an L among the distractor letters F in the given trial. At the beginning of the experiment, they were explicitly instructed to pay attention to the cue as it would correctly predict the location of the target in most, but not all of the trials. Observers were further instructed to continuously fixate at the fixation dot in the middle of the screen. The size of the OBA effect has been found fairly independent of the predictability of the cue in previous studies: similar effect sizes were reported with fully random[35] and highly predictable cues[32]. Therefore, the accuracy of the cue in the present study was set to 55%, which was estimated to be sufficient to elicit the OBA effect.

Each CBA block consisted of 144 trial scenes with 80 valid-cue trials (i.e., the cue appeared at the same location as the target), and 64 invalid-cue trials. Of the 64 invalid-cue trials, the target appeared on the cued chunk in 24 trials (Fig. 3a, right inset), whereas in the other 24 trials, the target appeared at the same distance from the cued location as in the first 24 trials on the uncued chunk (Fig. 3a, left inset). The remaining 16 invalid-cue trials were used for balancing the frequency of the individual shapes across the block and used only one chunk and two individual shapes in the scene, with the cue appearing in one of the individual shapes. These trials were not used in the subsequent analysis. The targets and the cues appeared randomly and the same number of times in all four locations of the 2-by-2 layout. In the invalid-cue trials, the target never appeared in the position diagonally opposite to the cued location.

*Object-based attention paradigm.* 49 participants completed 4 blocks of classic OBA task at the end of the experiment and data of 44/49 observers were analyzed (see exclusion criteria in "Participants" section). In the OBA blocks, the task was identical to the task in the CBA paradigm, but the target and distractor letters appeared in objects defined by visual boundary cues (i.e., rectangles or squares) instead of the shapes (Fig. 3b, OBA panel). We used the boundary-outlined rectangles as objects following previous studies[32–36] and augmented those with squares as analogs of the individual shapes constituting the diagonal pairs in the CBA paradigm. Observers completed the same number of trials of the same trial types (either two rectangles-comparable to trials with two chunks- or one rectangle and two squares-comparable to trials with one chunk and two individual shapes) with the same cues, and target locations as in the CBA blocks in a different random order.

*Familiarity test.* The Familiarity test was identical to the test in Experiment 1 with one modification driven by the goal of increasing the number of trials for a more accurate estimate of learning performance while keeping the appearance frequency of

the shapes and pairs balanced. Specifically, we introduced foil pairs and catch trials in this test in the following manner (see Supplementary material for more information on foil pairs). Observers performed 24 trials in which all true pairs were tested against foil pairs. In each trial, the true and foil pairs contained different shapes. From the 24 trials, 16 were normal and 8 were catch trials. In the catch trials, observers had to compare two foil pairs. These trials were needed to keep the appearance frequency of the shapes and pairs equal in the Familiarity test. In this way, both the true and the foil pairs appeared four times, and each shape appeared eight times in the test. The presentation order of the trials, and the sequential order of the true and foil pairs in a trial were separately randomized for each subject.

*Data analysis*. In all statistical analyses, we performed the standard two-sided frequentist and the corresponding Bayesian tests and drew our conclusion based on both types of tests combined. In the reported results, the value of the Bayes Factor directly reflected how much more probable the alternative hypothesis was compared to the null hypothesis. For computing the Bayes Factor, we used JZS Bayes factor analysis with a scaling factor of $\sqrt{1/2}$ in the Cauchy prior distribution[37–39].

**Reporting summary**. Further information on research design is available in the Nature Research Reporting Summary linked to this article.

## Data availability

The source data of the figures are provided with this paper and the data set generated during the current study is publicly available in the statistical-chunk-based-attention GitHub repository, https://github.com/GaborLengyel/statistical-chunk-based-attention. Source data are provided with this paper.

## Code availability

The code implementing the analyses in the current study is publicly available in the statistical-chunk-based-attention GitHub repository, https://github.com/GaborLengyel/statistical-chunk-based-attention.

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

## Acknowledgements

This project has received funding from the European Research Council (ERC) under the European Union's Horizon 2020 research and innovation program (project: COGTOM, GA No. 726090), and from the Office of Naval Research (N62909-19-1-2029).

## Author contributions

G.L., M.N., and J.F. designed the study. G.L. and M.N. performed experimental studies. G.L. analyzed the experimental data. All authors interpreted the results. G.L. and J.F. wrote the paper, with comments from M.N.

## Competing interests

The authors declare no competing interests.
