## [Peer Review File · Nature Communications]

Reviewers' Comments:

Reviewer #1:

Remarks to the Author:

This is an interesting study with meaningful results. While the authors' intention was to study the genesis of object representation, I believe the design and results allow them to make statements about the consequence of visual statistical learning, especially related to stimulus chunking and observers' attentional distribution.

The basic finding is that following visual statistical learning, the "a set of newly learned arbitrary statistical contingencies" manifested similar kinds of object-based effects in visual search and attentional cueing as true objects did. Logically, however, even though learned statistical contingencies lead to effects similar to what could be generated from objects, unless such effects are EXCLUSIVELY object-based, then results from the current study could not be interpreted as object representations were formed by learned statistical contingencies. In other words, the chunks defined by stable statistical contingencies could induce perceptual effects similar to the object-based effects, but that doesn't mean the chunks qualify as newly learned objects.

For example, the learned statistical contingencies could simply help guiding observers' attention, and in turn give rise to the effects observed in the visual search and attentional cueing experiments.

A technical point: there is substantial variability in observers' familiarity performance, with some performed well above chance. Given this, it is not accurate to claim that the statistical learning was implicit. Indeed, the main search and attentional effects came from observers showing higher level of familiarity performance.

Reviewer #2:

Remarks to the Author:

This paper reports 4 experiments (somewhat confusingly labeled 1, 1b, 1c, and 2) demonstrating that statistically-defined visual objects enjoy some of the same benefits as "real" objects such as the same-object cuing benefit. This provides further support for the idea that human observers are capable of forming meaningful "chunks" out of properties that tend to co-occur together, something first hypothesized by Horace Barlow (e.g. "predictive associations", see Vision Research 1990, but going back to the 1960s). I found these results pretty interesting and mostly convincing, although I think the discussion could be improved as described below. The results will be important to the community of statistical learning and perceptual grouping researchers, though perhaps not completely unsurprising since this finding is essentially what you would expect if statistically-defined objects are learned in the manner that statistical learning researchers have long claimed.

To anticipate an objection I do not share, some readers may be put off by the fact that the authors did not consistently get the kind of object benefits that the literature might lead one to expect - typically they occurred in only the first block and then decayed. However in my own lab we have found these results to be much more difficult to replicate than one would expect from the rhetoric in some papers. So I was pretty satisfied that they did get such effects in parallel conditions with statistical objects and "real" objects as used in previous studies.

I have a philosophical objection to the authors' use of the term "real objects" (in some places "true" objects) to indicate objects induced by ordinary processes of perceptual grouping. These objects are no more "real" than the statistically-defined ones. I would simply call them "geometrically defined objects" or "configural objects"

In connection with this terminological point, I think it would improve the discussion if the authors

would discuss the fact that the distinction between “statistically-defined objects” and “geometrically-defined objects” breaks down a bit if one views geometric object hypotheses as probabilistic hypotheses, as in some modern perceptual grouping theory (e.g. Froyen et al., Psych Review, 2015). Ordinary boundary-defined objects are certainly not created through the same mechanisms as those created by pairwise element contingencies, but both can be regarded as probabilistic hypotheses developed through some kind of probabilistic inference process. Along these lines the recent paper of Siegelman et al. (Cognition, 2019) takes a step to formalizing statistical learning as a Bayesian process and helps bridge this gap. I don’t think there is space in the current paper to develop this connection in depth, but it would help clarify the theoretical connections if the authors could briefly allude to this issue. It would also help tremendously if the strength of statistical objects hypotheses were modeled via some mathematics, which the SL literature rarely seems to try (though see the Siegelman paper), but that is probably too ambitious for this paper.

The authors use Bayes Factors side-by-side with frequentists tests throughout, but do not seem to pay attention to the BFs. For example in several places (eg p13, twice in the top paragraph) a BF of 3 given alongside a significant finding, and once described as “substantial evidence” of a difference. But a BF of 3 is pretty weak, at best (in Jeffrey’s famous breakdown) a “moderate” amount of evidence. I personally believe the BFs more than the p’s, so I think the authors need to go through and at least acknowledge the points in the argument where the Bayesian analyses do not strongly support their hypotheses.

Also standard practice is to give the BFs with an explicit direction, eg $BF_{10} = 3$, not just $BF = 3$ which leaves it unclear whether the alternative or the null is in the numerator.

Minor points:

It is somewhat confusing to call the second experiment 1b, when there was no 1a. At the very least call them Exp 1a, 1b, 1c, and 2. But I don’t see why they aren’t 1, 2, 3, and 4.

p11 less error -> fewer errors

p16 Fix Schofield in the bibtex (use consistent middle names etc).

p17 line 6 - this sentence is somehow messed up (ungrammatical and hard to understand)

Several places -- the authors tend to use commas ungrammatically. For example

p18 “...and the amount of learning thus, we...” should be “and the amount of learning, so we...”

p22 “Data of observers, who mentioned” should be “Data of observers who mentioned...”

There are other examples.

Reviewer #3:

Remarks to the Author:

In this manuscript, the authors examined whether observers would find the target faster if it appeared on a shape within a statistical chunk than on a shape on a different chunk. The idea was interesting and worthy of investigation. However, given the results in two experiments, it was hard to conclude that statistical chunks resulted in object-based attention. The paper also fits better as a brief report in a disciplinary journal within vision science, rather than a wider scientific audience. Major problems are outlined below.

Major issues

1. In both experiments, the search scenes contained either two true pairs, or one true pair and two

random shapes. The results thus could be driven by greater attention to a particular pair during search, or greater attention to the true pair than the random shapes. In other words, the chunk itself didn't lead to object-based attention, but rather the presence of a chunk drew attention during the search task. This, theoretically, is the fatal flaw given the design of the search scenes.

2. Experiment 1, from Figure 2 it seems that the across-within differences disappeared in the 3rd and 4th blocks for the chunks experiments but persisted for the objects experiment. This is a huge discrepancy between the chunks and objects comparison, presenting weak evidence for the object-based attention argument for the chunks. In fact, the difference was only present in the 1st block for the chunks experiment, which raised the possibility that the result could be a fluke.

3. The correlation results in Experiment 1 (Fig 2) may again due to noise since the sample size in both experiments was small. It was also confusing that the majority of observers who scored $\leq 50\%$ on the familiarity test still showed a positive across-within difference. If observers did not show learning of the shapes, how could they show biases for the within-chunk targets? This further implies that the RT or accuracy results could be due to noise.

4. In Experiment 1, observers did not perform above chance at the familiarity test, suggesting that they were not able to identify the true pair in the test, which means a failure in learning the chunks. However, the authors insisted that the observers implicitly learned the chunks. What is the empirical basis for implicit learning? There was zero evidence in the manuscript to suggest successful implicit learning. Without learning, how could the authors draw conclusions on the chunk-based attention *as a result* of learning? This is baseless.

5. In Experiment 2, Figure 4, panel C, this is the most important panel in this figure. However, the uncued-cued RT difference seemed significantly weaker for chunks than for objects, again highlighting the differences between chunks and objects. This, combined with Fig 2 in Exp 1, shows that chunks may be processed differently than objects, refuting the claim that the shape pairs produce chunk-based attention, similar to object-based attention.

Minor issues

6. How were the sample sizes in Experiment 1 determined?

7. Please remove discussions on the brain since the experiments provided only behavioral data; discussions on brain functions are too far-fetched.

Rebuttal letter:

Reviewer #1 (Remarks to the Author):

This is an interesting study with meaningful results. While the authors' intention was to study the genesis of object representation, I believe the the design and results allow them to make statements about the consequence of visual statistical learning, especially related to stimulus chunking and observers' attentional distribution.

The basic finding is that following visual statistic learning, the “a set of newly learned arbitrary statistical contingencies” manifested similar kinds of object-based effects in visual search and attentional cueing as true objects did. Logically, however, even though learned statistical contingencies lead to effects similar to what could be generated from objects, unless such effects are EXCLUSIVELY object-based, then results from the current study could not be interpreted as object representations were formed by learned statistical contingencies. In other words, the chunks defined by stable statistical contingencies could induce perceptual effects similar to the object-based effects, but that doesn't mean the chunks qualify as newly learned objects.

We thank the reviewer to raise this point as it is related to THE quintessential message of our work, and the reviewer’s comment amply demonstrates the fundamental importance of the two issues we jointly investigate in the present manuscript: what is an object, and is objectness an all-or-none feature? The answers we promote here are that a) object IS a sufficient set of statistical contingencies, and b) therefore, objectness is defined on a CONTINUOUS scale rather than in a binary either/or way despite the fact that this distribution in natural settings is heavily bimodal with a strong sense of objectness or non-objectness.

If our answers are correct, the reviewer’s logical criticism does not hold since it is not clarified at what point the induced similar perceptual effects suffice to call the representation a “newly learned object”. In other words, if objectness can be achieved in multitude ways based on different sets of statistical contingencies and these contingencies provide various levels of strength for objectness, it is impossible to ask for “exclusively” object-based effects for three reasons. First, it is not established that a particular “object-based” effect can be evoked only by objects. Second, the full set of effects defining an object does not exist. Third, any one effect would be evoked even by two “true” objects to a different extent, leaving open the question whether they produced the same or just similar effect. Hence, the best we can do is to show that the “chunk defined by stable statistical contingencies” can evoke an effect albeit to a lesser degree that originally was considered purely object-based.

To appreciate our point it is crucial to realize that it is impossible to design a paradigm that would show exclusively “object-based” effects simply because an accepted definition of an “object” is missing in the literature. Almost all “object-based” experiments in the field are done with abstract indicators of an object (e.g. a rectangular drawn on a paper or display) rather than with real objects. Therefore, they need to be considered as experiments where the existence of an object is “suggested” by indicators, i. e. where sufficient evidence is provided that an object should be there. Our experiments squarely fit in this tradition - we provide evidence in terms of statistical contingencies that an object should be there.

The novelty of our approach is that we provide this evidence in a manner that deliberately avoids traditional cues of objectness, such as contours. What cues are necessary and/or sufficient in general to identify some sensory input as an object is a subject of intensive debate in the literature (e.g. Kellman and Spelke, 1983; Spelke, 1990; Palmer & Rock, 1994). According to previous studies, stable boundaries such as luminance contours are the strongest criteria for “objectness” (Kellman and Spelke, 1983; Spelke, 1990; Palmer & Rock, 1994), and this is the reason why studies investigating object-based perceptual effects typically used clear luminance contours to define objects (Egley et al., 1994; Lee, Mozer, Kramer, & Vecera, 2012; Moore, Yantis, & Vaughan, 1998; Shomstein & Yantis, 2004; Vecera, 1994). It is worth noticing that such contours are nothing else but strong statistical contingencies of luminance edge segments appearing and moving in statistical coherence. However, there are several studies suggesting that having stable boundaries based on luminance discontinuities is not a necessary criterion for objectness as objects can be defined by texture-, disparity-, regularity-, symmetry-, or motion-based “boundaries” (e.g. Feldman, 1997, Schofield, 2000, Julesz, 1971). This is exactly the starting point of our research, asking whether there is a more general principle that could be used for defining objectness in these many different scenarios. We propose that the principle of statistical coherence might be the more general rule that could explain perception of objectness in those paradigms. Therefore, the emerging chunks DO qualify as newly learned objects, since they are defined by boundaries based on co-occurrences, which is agreement with all other object-definitions relying on boundaries based on some contingencies of some statistical properties.

We are fully aware of the fact that the “newly learned objects” we created by our statistical learning method are lacking several features of real mental objects based on a large set of more typical cues and statistics that emerge over years of interaction with the environment. This is the reason why we call our chunks object-LIKE representation. These object-like representations probably need to pass a number additional criteria to become real mental objects with the full richness of those mental constructs. It will be for future studies to investigate whether and how statistical learning mechanisms can by itself produce real mental objects or what additional processes are needed. We only making

the first step here by claiming that conceptually, there is a close link between chunking based on statistical learning and object segmentation based on visual boundary cues: both kinds of segmentation rely on statistical coherence, they both serve as fundamental components in forming object representations, and they are both sufficient to elicit some object-based perceptual effects. We clarified this message in the text and modified the introduction (lines 41-44, 67-70, 101-107) and the discussion (479-484, 537-542).

For example, the learned statistical contingencies could simply help guiding observers' attention, and in turn give rise to the effects observed in the visual search and attentional cueing experiments.

This is an important point, and we thank the reviewer to raise it. The same way as we can talk about different levels of objectness, we can also talk about (at least) two levels of attention modulation. The first is when the observer learns some latent structure in the input (i.e. chunks) and the very existence of such a structure in a scene automatically enhances spatial attention at the location where the chunk is presented. An example of such effect on attention has been reported by Zhao et al. (2013, *Psyc Sci*) showing that when one of four locations visual patterns were presented such that presentation in only one of the locations followed a temporal statistical structure, attention was enhanced at the location with structure. The second, more advanced level of attention modulation is when there are, say, two chunks in the scene, thus a priori neither of the two locations has an attentional advantage over the other when a target appears. However, when a preceding cue is provided that tags one PART of one of the chunks, OTHER PARTS of the same chunk benefit from the cueing so that in subsequent appearance of the target, it enjoys the attentional effect only if the target stays within the cued chunk. This latter type of effect is what would be equivalent to the traditional object-based attention and this is the effect we are after in our manuscript.

The reviewer raised the point that we need to show explicitly that our results cannot be explained simply by the first level of attention modulation in order to claim object-based attention effects. To accommodate the reviewer's point, we performed an additional analysis following exactly the reasoning above. In both experiments in the search tasks, we only considered half of the trials, in which two true-pairs (statistical chunks) appeared in the scene. Thus, in these trials, the attentional effect could not be explained by an increased attention towards the statistical chunks (true-pairs) because in both the within and across chunks conditions the targets appear on a true-pair (statistical chunk). Therefore, the effect could be caused only by an advantage of within-chunk processing over across-chunk processing similar to the advantage of the within-object processing over the across-object processing. We found the same chunk-based effect in these trials containing only true-pairs (Exp1: $t=3.77$, $p<0.001$, $BF=64$; Exp2 $t=2.43$, $p=0.02$, $BF=2$) as

in the full set of trials. The t and BF values are smaller in these analyses due to the increased variance which was the consequence of using only half of the trials, but the size of the effect remained almost the same (Exp1: $d_{\text{all trials}}=0.64$, $d_{\text{only pairs}}=0.49$; Exp2: $d_{\text{all trials}}=0.27$, $d_{\text{only pairs}}=0.26$) and the effects remained significant. We included these analyses in the result section of the revised manuscript (lines 240-251, 406-413, 1081-1100).

A technical point: there is substantial variability in observers' familiarity performance, with some performed well above chance. Given this, it is not accurate to claim that the statistical learning was implicit. Indeed, the main search and attentional effects came from observers showing higher level of familiarity performance.

There are two issues intertwined here: explicitness of knowledge and the size of effect demonstrated by the individual observers. Regarding the issue of explicitness, we have excluded all participants with explicit knowledge from all of our analysis because we were interested in the effects of implicit automatic processes (that could potentially be related to object representation) and not the consequence of explicit cognitive knowledge and strategies. Therefore, we determined explicit knowledge at the end of the experiments during the debriefing. Data of observers, who mentioned anything about noticing consistent pairs during the experiments was excluded from all the analyses we performed in the study. This was explained in the method section (lines 576-584, 660-665, 683-694), but we failed to mention in the main text - we thank the reviewer to catch this. We modified the main text at lines 127-129, 334-337 to include this crucial information.

Regarding the second issue, the size of effect shown by individual observers, it is expected that observers who performed better in the familiarity test, i. e. who developed a stronger sense of “objectness” based on the learned statistical contingencies would exhibit a stronger attention effect. Therefore, the observation that “higher effect comes from better learners” is exactly in line with our argument saying that learning statistical contingencies is responsible for the attentional effect.

Reviewer #2 (Remarks to the Author):

This paper reports 4 experiments (somewhat confusingly labeled 1, 1b, 1c, and 2) demonstrating that statistically-defined visual objects enjoy some of the same benefits as “real” objects such as the same-object cuing benefit. This provides further support for the idea that human observers are capable of forming meaningful “chunks” out of properties that tend to co-occur together, something first hypothesized by Horace Barlow (e.g.

“predictive associations”, see Vision Research 1990, but going back to the 1960s). I found these results pretty interesting and mostly convincing, although I think the discussion could be improved as described below. The results will be important to the community of statistical learning and perceptual grouping researchers, though perhaps not completely unsurprising since this finding is essentially what you would expect if statistically-defined objects are learned in the manner that statistical learning researchers have long claimed.

We thank the reviewer for these kind words. We agree that the idea of chunks and objects emerging from learning statistical contingencies has been implicitly advocated by Barlow as we prominently mentioned it in the original paper on visual statistical learning (Fiser & Aslin 2001 *Psyc. Sci*). However, we would also like to emphasize that the full-fledged argument making this link explicit by showing object-based perceptual effects with statistically defined chunks is an original contribution of the present study. It provides much missing evidence for the claim that statistical learning is intimately related to object representations, which might be expected by the reviewer, but is certainly not widely accepted yet as demonstrated by the comments of Reviewer 1.

To anticipate an objection I do not share, some readers may be put off by the fact that the authors did not consistently get the kind of object benefits that the literature might lead one to expect - typically they occurred in only the first block and then decayed. However in my own lab we have found these results to be much more difficult to replicate than one would expect from the rhetoric in some papers. So I was pretty satisfied that they did get such effects in parallel conditions with statistical objects and “real” objects as used in previous studies.

We agree with the reviewer and this was one of the reasons to run the experiments with objects defined by visual boundaries and with chunks defined by statistical coherence in parallel. We think the full pattern of results with chunks and objects support a very consistent interpretation, which we described at lines 257-290 and 339-393.

I have a philosophical objection to the authors’ use of the term “real objects” (in some places “true” objects) to indicate objects induced by ordinary processes of perceptual grouping. These objects are no more “real” than the statistically-defined ones. I would simply call them “geometrically defined objects” or “configural objects”

We agree with the reviewer’s comment. This is exactly the argument we spelled out in reply to Reviewer 1 - all previous experiments use geometric “indicators” of objects rather than true objects. However, these previous studies of object cognition made conclusions

based on their results about “real objects”, mostly based on the reasoning that those geometric objects pass one of the most important criteria for “objectness”, the “cohesion”, i.e. stable boundaries (Kellman and Spelke, 1983; Spelke, 1990; Palmer & Rock, 1994) and thus, they are sufficient to elicit “objectness” in our brain. To clarify this situation, we changed the “real” and “true” to “objects defined by visual boundaries” throughout the text.

In connection with this terminological point, I think it would improve the discussion if the authors would discuss the fact that the distinction between “statistically-defined objects” and “geometrically-defined objects” breaks down a bit if one views geometric object hypotheses as probabilistic hypotheses, as in some modern perceptual grouping theory (e.g. Froyen et al., Psych Review, 2015). Ordinary boundary-defined objects are certainly not created through the same mechanisms as those created by pairwise element contingencies, but both can be regarded as probabilistic hypotheses developed through some kind of probabilistic inference process. Along these lines the recent paper of Siegelman et al. (Cognition, 2019) takes a step to formalizing statistical learning as a Bayesian process and helps bridge this gap. I don’t think there is space in the current paper to develop this connection in depth, but it would help clarify the theoretical connections if the authors could briefly allude to this issue. It would also help tremendously if the strength of statistical objects hypotheses were modeled via some mathematics, which the SL literature rarely seems to try (though see the Siegelman paper), but that is probably too ambitious for this paper.

There are several points here to elaborate. First, we strongly agree with the reviewer that a probabilistic computational framework is the right approach to handle the phenomena investigated in our manuscript together with grouping and Gestalt effects. We thank the reviewer for calling our attention to the Froyen et al. (2015) reference, which we added to the section of our manuscript discussing common computational frameworks for object cognition and statistical learning (lines 551-555). We would also agree that, while on Marr’s implementation level (a. k. a. recognition model) ordinary boundary-defined objects are created through a number of specialised mechanisms, which our “chunk-objects” defined by pairwise element contingencies use far less, computationally, the emergence of the two types of object would be naturally handled by the same inferential model. In fact, this is the main message of our paper.

Second, as a matter of fact, we do have a mathematical model that explains SL in a probabilistic framework, which uses the “strength of statistical objects hypotheses” the reviewer asks for (Orban et al, 2008, PNAS). This model demonstrates how an ideal learner creates a latent representation of the chunks and how an inferential procedure on this representation replicates human experimental results to an impressive degree. We

see the Orban model as the appropriate basis by a direct generalization to make the link between such probabilistic representation and object-based perceptual effects. However, we also agree with the reviewer that developing this probabilistic model is a formidable work which goes beyond the scope of the present manuscript. We expanded our ideas a bit more in the discussion about possible computational mechanisms of object representations and their use.

Third, while the Siegelman et al. (2019) paper is a good example of applying Bayesian data analysis, it is not a probabilistic model explaining mechanism of SL, a fact that the authors themselves acknowledge early in the discussion of their paper (page 7, General discussion, first sentence in the second paragraph). The paper shows how a hierarchical Bayesian model of the data can help testing sophisticated hypotheses about different learning strategies within subjects. Thus, while the Siegelman et al. (2019) paper does a good service in spreading the Bayesian thinking, appropriate probabilistic models of SL should be sought elsewhere (see for example Fiser & Lengyel 2019 Curr. Opin in Neurobi.)

The authors use Bayes Factors side-by-side with frequentists tests throughout, but do not seem to pay attention to the BFs. For example in several places (eg p13, twice in the top paragraph) a BF of 3 given alongside a significant finding, and once described as “substantial evidence” of a difference. But a BF of 3 is pretty weak, at best (in Jeffrey’s famous breakdown) a “moderate” amount of evidence. I personally believe the BFs more than the p’s, so I think the authors need to go through and at least acknowledge the points in the argument where the Bayesian analyses do not strongly support their hypotheses.

We agree with the reviewer that BFs are more informative measures of importance than significance values, therefore, we do pay attention to our BFs. However, we respectfully disagree with the reviewer’s observation that our use of BFs is inadequate. One of the advantages of BF is that its value quantifies the weight of evidence for the hypotheses in a continuous manner anchored on the value of 1, i.e. zero evidence supporting either hypothesis. Therefore, any hard division at various thresholds is an arbitrary decision serving convenience in communication. In most arbitrary categorizations BF~3 is chosen as the lower bound of the substantial evidence category (Jeffreys & Harold (1998) *The Theory of Probability*, Oxford; Kaas & Raftery (1995) *Jou of Am. Statist. Assoc.*). Even in Jeffrey’s original breakdown this lower limit is close to 3, and it is 3.16 only because it is derived from the log10 scale as $10^{0.5}$. In contrast to the significance p value, where comparing $p < 0.049$ vs. $p < 0.051$ is uninterpretable (regarding the probability of our hypotheses), BF = 2.9 and BF= 3 are interpretable as providing quite similar amounts of favourable evidence for the given hypothesis. Therefore, while we acknowledge that BF = 3 is at the low end of the category of “substantial evidence”, it is nevertheless an

accepted value for claiming a true effect by itself. Moreover, throughout the manuscript, we always use the BF in combination with the frequentist significance value of p in our analyses, and our claims of “substantial evidence” always refer to the assessment of the combined result of significant p value and BF around or above 3. The only place where we use only BF alone for claiming substantial evidence is when data from Experiments 1a and 1b are pulled, and thus it is inappropriate to use simple frequentist statistics and using BF is more intuitive than relying on complicated frequentist meta-analysis and other correction methods. However, in this case we obtained the formidable value of $BF = 2907$, which is way beyond decisive evidence by any arbitrary tabulation. We included a paragraph in the Method section to clarify these points (774-780).

Also standard practice is to give the BFs with an explicit direction, eg $BF_{10} = 3$, not just $BF = 3$ which leaves it unclear whether the alternative or the null is in the numerator.

We agree that the shorthand of $BF = 3$ is misleading without further specification of the supported hypothesis. We made corrections in the manuscript.

Minor points:

It is somewhat confusing to call the second experiment 1b, when there was no 1a. At the very least call them Exp 1a, 1b, 1c, and 2. But I don't see why they aren't 1, 2, 3, and 4.

We agree with the reviewer and we relabeled experiment 1 to experiment 1a so that it would be consistent with experiments 1b and 1c. The numbering of the experiments reflects the fact that we used the same paradigm in experiments 1a, 1b, and 1c and an entirely different paradigm in experiment 2. Therefore, we would like to keep the 1a, 1b, 1c, 2 numbering to emphasize the distinction between the two paradigms.

p11 less error -> fewer errors

Corrected.

p16 Fix Schofield in the bibtex (use consistent middle names etc).

Fixed.

p17 line 6 - this sentence is somehow messed up (ungrammatical and hard to understand)

Rephrased (line 496-498).

Several places -- the authors tend to use commas ungrammatically. For example

p18 “...and the amount of learning thus, we...” should be “and the amount of learning, so we...”

p22 “Data of observers, who mentioned” should be “Data of observers who mentioned...”
There are other examples.

We corrected these and other grammatical errors in the manuscript.

Reviewer #3 (Remarks to the Author):

In this manuscript, the authors examined whether observers would find the target faster if it appeared on a shape within a statistical chunk than on a shape on a different chunk. The idea was interesting and worthy of investigation. However, given the results in two experiments, it was hard to conclude that statistical chunks resulted in object-based attention. The paper also fits better as a brief report in a disciplinary journal within vision science, rather than a wider scientific audience. Major problems are outlined below.

Major issues

1. In both experiments, the search scenes contained either two true pairs, or one true pair and two random shapes. The results thus could be driven by greater attention to a particular pair during search, or greater attention to the true pair than the random shapes. In other words, the chunk itself didn't lead to object-based attention, but rather the presence of a chunk drew attention during the search task. This, theoretically, is the fatal flaw given the design of the search scenes.

This is the same point raised by Reviewer 1, and we addressed the concern in detail in our reply there. Briefly recapitulating the answer here, the point raised by the review is not a fatal flaw, since the scenes were designed so that we could rule out the concern the reviewer raised. Specifically, in all of our experiments, we had trials in the search task that contained only two true pairs without individual shapes. In these trials, more attention to true pairs could not lead to the attention effect since the shapes in both the cued and uncued chunk were a part of a true-pair. Nevertheless, when we performed the same analysis on these trials only, we obtained the same object-based attention as with the entire set of trials (Exp1: $t=3.77$, $p<0.001$, $BF=64$; Exp2 $t=2.43$, $p=0.02$, $BF=2$, see lines 240-251, 407-414, 1082-1101 in the new manuscript). Therefore, the effect qualifies as an object-based attentional effect.

2. Experiment 1, from Figure 2 it seems that the across-within differences disappeared in the 3rd and 4th blocks for the chunks experiments but persisted for the objects experiment. This is a huge discrepancy between the chunks and objects comparison, presenting weak evidence for the object-based attention argument for the chunks. In fact,

the difference was only present in the 1st block for the chunks experiment, which raised the possibility that the result could be a fluke.

We respectfully disagree with the reviewer's assessment for three reasons. First, we replicated all the significant results of Experiment 1 in Experiments 1B so it is, at a minimum, odd to call those results a "fluke". Second, and more importantly, the Bayes factor supporting the chunk-based error rate effect was 2907. This is exceedingly strong evidence indicating that our results were not a "fluke". The Bayes Factor for the positive correlation between the familiarity performance and the error rate effect was 24, which is again very strong statistical evidence that the effect was not noise but a true effect of learning. These are strong quantitative indicators of our effects that cannot be simply dismissed. Third, we were specifically cautious about the fact that the effect disappeared in Blocks 3 and 4 for chunks, this is why we conducted the identical control experiment 1c with objects. We gave a detailed explanation in the manuscript as to how the combination of two components, 1) the expected -and statistically confirmed- weaker effect of chunks compared to those with objects, and 2) the general diminishing tendency of the effect for both chunks and objects across blocks due to learning led to the insignificance of the effect in later blocks for chunks. Our strong positive statistical results together with the quantitative treatment of the diminishing nature of the effect over time provides a parsimonious explanation of our findings. This view was echoed by Reviewer 2's comments. We are at loss why the reviewer did not comment at all on these results and explanation of ours presented in the manuscript. We hope that the new version of the manuscript reflects these points more clearly to avoid further confusion (lines 277-290).

3. The correlation results in Experiment 1 (Fig 2) may again due to noise since the sample size in both experiments was small. It was also confusing that the majority of observers who scored $\leq 50\%$ on the familiarity test still showed a positive across-within difference. If observers did not show learning of the shapes, how could they show biases for the within-chunk targets? This further implies that the RT or accuracy results could be due to noise.

As with the attention effects, we provided very strong evidence for a positive correlation between the chunk-based error rate effect and the performance on the familiarity test in Experiment 1, so the results could not be interpreted as noise due to small sample size. First, we replicated the significant positive correlation of Exp. 1 in Exp 1b. Second, we obtained $BF=24$, which indicates a very strong support for the existence of the correlation. Third, we included a partial correlation analysis to rule out the possibility that the correlation between the chunk-based error rate effect and familiarity we reported was just due to a generic factor such as attention or across-subject variability in overall

performance (lines 230-240). This analysis confirmed the existence of independent correlation between the attention effect and familiarity. In the new manuscript, we added a fourth measure based on trials where only two chunks were presented, and showed specifically that the object (chunk)-based attention component of the effect is also significant (lines 240-251). This set of strong statistical results clearly suggests that the positive relationship between the chunk-based effect and the performance in the familiarity test is a real effect based on sufficient data and not an accident caused by random noise.

Regarding the $\leq 50\%$ performances, performing strongly below chance on the familiarity test does not mean that these observers did not learn anything about the structure of scenes (and not of the shapes as the reviewer says!) since that would create chance performance. Instead, it means that the observers learned quite a bit, and they used this knowledge in a way that prompted them to prefer the random pairs, not the true pairs in the familiarity test. This is related to the well-known familiarity/novelty dichotomy in preference documented in the literature. Nevertheless, the effect of this knowledge can still fully serve object-based attention as it is clear from the fact that the object-based effect for the majority of observers who performed below 50% in the familiarity test is positive. Had the number of people with novelty and familiarity preference been the same, the correlation between the object-based effect and learning would have taken a U-shaped quadratic form. Since familiarity preference is typically stronger in these experiments, despite the observers with below chance familiarity performance, we factually demonstrated a very strong overall positive linear correlation between the magnitudes of learning and the attention effect as explained above. In any case, these results cannot be parsimoniously explained by the assumption of measuring noise. We are open to any argument challenging this statement based on valid statistical reasoning.

4. In Experiment 1, observers did not perform above chance at the familiarity test, suggesting that they were not able to identify the true pair in the test, which means a failure in learning the chunks. However, the authors insisted that the observers implicitly learned the chunks. What is the empirical basis for implicit learning? There was zero evidence in the manuscript to suggest successful implicit learning. Without learning, how could the authors draw conclusions on the chunk-based attention *as a result* of learning? This is baseless.

The lack of significant performance in the familiarity test might seem puzzling at first sight given that participants must have represented the chunks to have the chunk-based effects as argued above. However, there are three reasons why participants' performance on the familiarity test in our study were smaller than in previous statistical learning studies. First, the interleaved visual search task blocks interfered with the Familiarity test which decreased the performance in the subsequent familiarity task. Second, the statistical

exposure scenes in one block were shorter (half as long) as in the classical studies (Fiser & Aslin 2001), and this further reduced the strength of statistical learning by the participants. Third, familiarity is a relatively crude summary measurement of learning as pointed out by several studies (Siegelman, Bogaerts & Frost, *Behavior Research Methods*, 2017), while the visual search task is a more sensitive task to measure effects of statistical learning, and this why the apparent discrepancy emerges.

Crucially, as we explained in our manuscript, a significant OVERALL level of familiarity is NOT a necessary prerequisite of demonstrating a correlation between object-based attention effects and learning (line 185-189). The necessary prerequisite is sufficient variability in learning performance among the observers so that correlation between learning and attentional effects could be reliably established. As it is well-known in statistics, means and variances are independent characteristics of a data set, and a perfect correlation can be achieved between two sets of which one or both have a mean of zero (in our case, average behavior at chance level). Applying this to our experiment, we obtained performances in the familiarity test ranging between chance and 90%, which -contrary to the reviewer's claim- means plenty of implicit learning, just the mean of this learning across observers happened not to deviate strongly from 50%. However, this learning provided more than a sufficient range for computing the overall correlation with the attention effect, which turned out to show a strong correlation ($R=0.4$ for both the main and the replication experiments), and a very strong evidence that this positive correlation is a real effect ($BF = 24$). Furthermore, we conducted one successful extra test in the original manuscript (partial correlation analysis on the full data set) and as second one in the new version (chunk-based error rate analysis on the data set restricted to scenes with only pairs) to reinforce the conclusion that the cause of the strong correlation is implicit chunk learning. We find it puzzling to call our conclusion relying on the results of this set of analyses as "baseless".

Finally, in Experiment 2 with a less intricate task design, we had provided results, where in addition to a sufficient range of learning, the overall level of learning also happened to be significantly above chance (Fig 4E). Therefore the reviewer's comment that "*There was zero evidence in the manuscript to suggest successful implicit learning.*" is incorrect even in the sense of average performance.

5. In Experiment 2, Figure 4, panel C, this is the most important panel in this figure. However, the uncued-cued RT difference seemed significantly weaker for chunks than for objects, again highlighting the differences between chunks and objects. This, combined with Fig 2 in Exp 1, shows that chunks may be processed differently than objects, refuting the claim that the shape pairs produce chunk-based attention, similar to object-based attention.

Since we are demonstrating how object-based effects emerge due to statistical contingencies, we used a setup in which there were NO contingencies between elements at the beginning, and demonstrated how implicit learning based on limited exposure facilitates the emergence of the effect due to encoding SOME contingencies. To justify our argument, we showed how similar these effects are to the ones manifested with geometrically defined objects and how much they were related to the amount of learning. These are proper characteristics of the effect that can be used for our argument. In contrast, it is unrealistic to request to have the same amplitude of effect across the two situations, one based on implicit learning in the last half an hour, the other based on lifelong learning with much definite explicit cues. Therefore, the inference that a weaker effect in the case of chunks is a valid indicator of different underlying processes is incorrect, and arguments based on amplitude comparison should not be used for judging our results. We clarified this point in the new manuscript (lines 375-377).

Minor issues

6. How were the sample sizes in Experiment 1 determined?

Since we used a novel paradigm in Exp 1. We could not do a statistical power analysis to determine the sample size. Therefore, we aimed to recruit 30-30 participants considering the sample sizes in previous statistical learning (e.g. Fiser & Aslin, 2001, 2005) and object-based processing studies (e.g. Baylis & Driver, 1993; Luck & Vogel, 1997; Vecera, Behrmann, & McGoldrick, 2000, Egly et al., 1994; Vecera, 1994).

7. Please remove discussions on the brain since the experiments provided only behavioral data; discussions on brain functions are too far-fetched.

We ask the reviewer to be more specific about the part s/he requests to be removed due to being “far-fetched”, as we are uncertain what the reviewer means by “discussion on the brain” given that we consider behavior, computation and neural implementations all parts of discussion about the brain. If the reviewer meant discussing neurophysiological results, we made a single reference to an earlier fMRI study providing evidence that visual error predictions spread only across parts of the same object, and a second reference clarifying the type of gamma synchrony we made a specific prediction about in the paper based on our results. We consider using both references adequate, but if these are the “discussions” in question, and the editor also thinks that those are far-fetched, we will remove them.

Reviewers' Comments:

Reviewer #1:

Remarks to the Author:

I appreciate that the authors put serious effort in addressing my concerns, however, the key issue related to the interpretation of their results remain unresolved.

I raised the concern that logically, "even though learned statistical contingencies lead to effects similar to what could be generated from objects, unless such effects are EXCLUSIVELY object-based, then results from the current study could not be interpreted as object representations were formed by learned statistical contingencies."

The authors' response to this question sounds circular to me. They defined that an "object is a sufficient set of statistical contingencies". If that's their definition of "object", then they should write an opinion paper to promote this definition of "object" and discuss all the advantages of this definition. This is different from claiming that their experimental evidence supports such a definition. The experimental results tell us that "a set of newly learned arbitrary statistical contingencies" can lead to certain attentional effects, namely they will attract attention as a chunk.

Their interpretation requires the demonstrated attentional effects to be "exclusively" object-based. Apples are sweet, but a sweet fruit is not necessarily apple, unless "sweetness" is an exclusive feature of apples. The authors stated in the response letter that "it is impossible to ask for "exclusively" object-based effects". Well, then they have a logical gap and should not draw the conclusion the way they did. They went on to say that "the best we can do is to show that the "chunk defined by stable statistical contingencies" can evoke an effect albeit to a lesser degree that originally was considered purely object-based." Ok, I'm not sure who considered the effect "purely" object-based. Even if some people "originally" considered the attentional effect purely object-based, isn't it more sensible to use the current results to revise the "original" misconception, so that we now understand the attentional effect is not "purely" object-based, that "chunks" can have similar attentional effects?

Regarding the issue of implicit vs. explicit effect, this hinges on ones' view on what the familiarity test reveal. I side with researchers (e.g., Turk-Browne, Seitz, Shams) who consider familiarity test as a measure of explicit learning. The fact that much of the effect come from observers who developed high level of familiarity with the learned pairs suggests explicit rather than implicit statistical learning. However, I understand that different researchers do hold different views on this point.

Reviewer #2:

Remarks to the Author:

This manuscript is much improved and I recommend publication. The results certainly clarify that status of statistically-defined objects in visual perception, though I can't say I found the results terribly surprising since a number of authors (including the authors in previous papers) have argued that all objects are probabilistically-defined. Still the new data certainly adds to the empirical support in favor of that claim, and will be cited for doing so.

Reviewer #3:

Remarks to the Author:

The authors have addressed my concerns.

Response to the reviewers' comments

Reviewer #1 (Remarks to the Author):

I appreciate that the authors put serious effort in addressing my concerns, however, the key issue related to the interpretation of their results remain unresolved.

I raised the concern that logically, “even though learned statistical contingencies lead to effects similar to what could be generated from objects, unless such effects are EXCLUSIVELY object-based, then results from the current study could not be interpreted as object representations were formed by learned statistical contingencies.”

The authors' response to this question sounds circular to me. They defined that an “object is a sufficient set of statistical contingencies”. If that's their definition of “object”, then they should write an opinion paper to promote this definition of “object” and discuss all the advantages of this definition. This is different from claiming that their experimental evidence supports such a definition. The experimental results tell us that “a set of newly learned arbitrary statistical contingencies” can lead to certain attentional effects, namely they will attract attention as a chunk.

Their interpretation requires the demonstrated attentional effects to be “exclusively” object-based. Apples are sweet, but a sweet fruit is not necessarily apple, unless "sweetness" is an exclusive feature of apples. The authors stated in the response letter that “it is impossible to ask for “exclusively” object-based effects”. Well, then they have a logical gap and should not draw the conclusion the way they did. They went on to say that “the best we can do is to show that the “chunk defined by stable statistical contingencies” can evoke an effect albeit to a lesser degree that originally was considered purely object-based.” Ok, I'm not sure who considered the effect “purely” object-based. Even if some people “originally” considered the attentional effect purely object-based, isn't it more sensible to use the current results to revise the “original” misconception, so that we now understand the attentional effect is not "purely" object-based, that “chunks” can have similar attentional effects?

We see the logic of the reviewer's argument claiming that the researchers conducting the original experiments coining the concept of “object-based attention” could have been more careful and instead use the phrase “attention evoked by 2D-drawing-of-objects”. Therefore, our findings suggests that the phenomenon is not “purely-object-based” but at least “object-and-chunk-based”. However, we dissent from the reviewer suggestion that this is the main message of our work which amounts to a renaming of the object-based-attention phenomenon. Furthermore, we disagree that our argument based on the proposal that objects should be viewed as a strong set of statistical contingencies is circular, and consequently, we maintain that asking for “exclusively object-based effects” is missing the more far-reaching main message of our work.

Specifically, our argument is not circular since it follows the standard method of hypothesis testing. First, we set the hypothesis that objects are a particularly strong aggregation of statistical contingencies as we know that classically defined object features (long contours, similar textures/colors, Gestalt structures, etc.) are all exemplars of such contingencies. Next, we kept such contingencies while removing the classical instantiations of these contingencies, and showed that the attentional effects attributed previously to objects still emerged. Moreover, we showed that the

strength of the attentional effect correlated with the strength of learning, thus the crucial factor of the effect appeared to be implicit access to the existing contingencies in the images. Based on this we concluded that the essential aspect that would drive these effects are most directly related to contingencies, whereas the classical features used to define objects are secondary manifestations of these contingencies. While we could have said that our method tapped into a different kind of effect that objects and contingency-based chunks share, we chose the more parsimonious path and suggested that, in line with our hypothesis, objects should be defined by these primary causes (contingencies). While this standard method of hypothesis testing actually yields a support for a working definition of objects, it is unclear what definition of objects and “exclusively object-based effects” the reviewer has in mind.

Reformulating the reviewer’s analogy with apples and sweetness, we propose that fruits (including apples) represent objects, sweetness stands in for object-based attention effect, and sucrose ($C_{12}H_{22}O_{11}$) is statistical contingencies. We know from previous research that apples are sweet, so previous research declared that sweetness comes from apples (also from pears, and oranges), so it is a fruit-based phenomenon. By associations, this would imply that experiencing sweetness, we need to taste round, relatively firm, palm-size food items. We showed in our experiment that cane-syrup (chunks) also evoke the taste of sweetness despite not having round (or any kind) of defined shape, being soft and liquid-like and not being fruit at all. Moreover, we showed that with our sugar-assay (measuring the amount of learning) we can predict well how much sweetness will be experienced, a prediction that previous fruit-based research could not provide. The reviewer suggests that we should correct the misconception and call the phenomenon fruit- and cane-syrup-based sweetness. We suggest that sweetness (object-based attention effect) is not directly related to fruits (objects) or syrups (chunks) per se but through sugar-content (statistical contingencies). Since fruits are sweet to different degree, sweetness is a good predictive feature of fruits, but the essence of fruits is to be nutritious for animals, so that they would eat the fruits and carry the seeds to long distances for effective dissemination. To understand this function, it is better to identify sucrose as a good instantiation of nutrition and fruits as items providing large nutrition value since this could be useful for generating future insights about fruits, syrups and biology rather than remaining at the surface description of sweetness. This is why we chose the basic feature of sucrose to define the essence of fruits.

We extended the discussion of the manuscript to reflect these point.

Regarding the issue of implicit vs. explicit effect, this hinges on ones’ view on what the familiarity test reveal. I side with researchers (e.g., Turk-Browne, Seitz, Shams) who consider familiarity test as a measure of explicit learning. The fact that much of the effect come from observers who developed high level of familiarity with the learned pairs suggests explicit rather than implicit statistical learning. However, I understand that different researchers do hold different views on this point.

We agree with the reviewer that declaring an effect to be implicit vs. explicit is a matter of how those categories are defined and what one assumes about the process of assessing familiarity. Nevertheless, it is important to keep different notions clearly separated. Explicit LEARNING in our vocabulary is a process where the observer has a well-defined task and tries to improve performance in that task. In this sense, our method is not using explicit learning, our observers had no task to perform beyond paying attention to the scenes. Explicit KNOWLEDGE is what can be achieved both by implicit and explicit learning, and indeed, a small fraction of our observers reached the explicit level with their knowledge after the familiarization part of the experiment. However, we excluded these from our analyses by testing for explicitness in the form of receiving a verbal report on

any hint that the observer became aware of the structure of the scenes. Thus, we suggest that our observers perform the test based for the most part on their implicit knowledge. On the other hand, report of familiarity can be used in arbitrary setups to assess the observers knowledge after either type of learning process and either type of acquired knowledge developed. We are not aware of any established study that can undoubtedly separate what components go into a familiarity judgement in general. Therefore, we think that in the main familiarity cannot be considered as a measurement of explicit learning or even explicit knowledge, and even less so in our case, when the observers have no task to perform and rely predominantly on their implicit knowledge.

We are confused by the sentence “*The fact that much of the effect come from observers who developed high level of familiarity with the learned pairs suggests explicit rather than implicit statistical learning.*” What effect the reviewer refers to in this context? The OBA effect in our manuscript, and specifically the correlation between familiarity and OBA? If so, we would point out again that the correlation exists even when the high familiarity participants’ data is removed from the analysis, and the fact that larger OBA is paired with stronger familiarity is a natural feature of correlations not to be confused with indicators of the strength of a correlation. Correlations are defined equally by points with high-high and low-low OBA-familiarity scores, thus it is not true that observers with high familiarity scores define the effect.

We extended the method section at the debriefing sections explaining how did we measured the explicitness of participants knowledge and some of the ideas above.

Reviewer #2 (Remarks to the Author):

This manuscript is much improved and I recommend publication. The results certainly clarify that status of statistically-defined objects in visual perception, though I can't say I found the results terribly surprising since a number of authors (including the authors in previous papers) have argued that all objects are probabilistically-defined. Still the new data certainly adds to the empirical support in favor of that claim, and will be cited for doing so.

Reviewer #3 (Remarks to the Author):

The authors have addressed my concerns.